# Selective inversion of rift basins in lithospheric-scale analogue experiments

Anindita Samsu[1,2], Weronika Gorczyk[3], Timothy C. Schmid[4], Peter G. Betts[2], Alexander R. Cruden[2], Eleanor Morton[2], Fatemeh Amirpoorsaeed[2]

[1]Institute of Earth Sciences, University of Lausanne, Lausanne, 1015, Switzerland
[2]School of Earth, Atmosphere and Environment, Monash University, Melbourne, 3800, Australia
[3]Centre for Exploration Targeting, School of Earth Sciences, University of Western Australia, Perth, 6009, Australia
[4]Institute of Geological Sciences, University of Bern, Bern, 3012, Switzerland

*Correspondence to*: Anindita Samsu (anindita.samsu@unil.ch)

**Abstract.** Basin inversion is commonly attributed to the reverse reactivation of basin-bounding normal faults. This association implies that basin uplift and inversion-related structures are mainly controlled by the frictional behaviour of pre-existing faults and associated damage zones. In this study, we use lithospheric-scale analogue experiments of orthogonal extension followed by shortening to explore how the flow behaviour of ductile layers underneath rift basins promote or suppress basin inversion. Our experiments show that the rheology of the ductile lower crust and lithospheric mantle, modulated by the imposed bulk strain rate, determine: (1) basin distribution in a wide rift setting and (2) strain accommodation by fault reactivation and basin uplift during subsequent shortening. When the ductile layers deform uniformly during extension (i.e., stretching) and shortening (i.e., thickening), all of the basins are inverted. When deformation in the ductile layers is localised during extension (i.e., necking) and shortening (i.e., folding), only some basins – which are evenly spaced apart – are inverted. We interpret the latter as selective basin inversion, which may be related to the superposition of crustal-scale and lithospheric-scale boudinage during the previous basin-forming extensional phase and/or folding of the ductile layers during shortening.

## 1 Introduction

Ancient rift basins record more than just the extensional event during which they formed. The initial basin-forming phase is commonly followed by subsequent events associated with thermal equilibration of the lithosphere (Morgan and Ramberg, 1987) or a change in the driving far-field plate kinematics (Forsyth and Uyeda, 1975). Some rifts fail before continental breakup and remain as fossil features within continents, which are likely to be overprinted by younger geological features. There are many examples from around the world in which the initial rift phase is interpreted to have been succeeded by shortening that resulted in basin inversion (Williams et al., 1989; Beauchamp et al., 1996; Turner and Williams, 2004; Blaikie et al., 2017; Le Gall et al., 2005; Elling et al., 2021; Thorwart et al., 2021). Sustained shortening (i.e., collision between two continental plates or blocks) can also form orogenic belts; the characteristics of these belts may record the influences of pre-existing extensional basins (e.g., NW Argentinian Andes, Carrera et al., 2006; Chungnam Basin, Park et al., 2019; Cape Fold Belt, Paton et al.,

2006). Modern examples of orogenic belts that were impacted by pre-existing basins include the European Alps and Apennines (Boutoux et al., 2014; Scisciani et al., 2019; Pace et al., 2022) and the Pyrenees (Mencos et al., 2015).

In this paper, we focus on "positive" inversion, which was defined by Williams et al. (1989) as the contraction of a region that previously underwent extension. Analogue modelling to date has focused on the role of crustal-scale extensional structures in accommodating strain during shortening, from the scale of the basin to that of individual basin-forming faults (e.g., Bonini et al., 2012; Molnar and Buiter, 2022; also see reviews by McClay, 1995 and Zwaan et al., 2022). Many analogue experiments on basin inversion have examined the influence of pre-existing normal faults or shear zones (e.g., McClay, 1989, 1995; Del Ventisette et al., 2006; Marques and Nogueira, 2008) and basin fill that is relatively weak compared to the extended crust (e.g., Panien et al., 2005) on deformation of the sedimentary layers within the basin. In these cases, specific assumptions are made on the behaviour of the viscously deforming crust and lithospheric mantle, and this behaviour is imposed as boundary conditions from the start of the experiments.

Complementary to analogue models, numerical experiments have focused on the drivers of basin inversion at the lithospheric-scale (e.g., Hansen and Nielsen, 2003; Sandiford et al., 2006; Buiter et al., 2009). They have examined the interactions between lithospheric-scale instabilities (e.g., necking), the thermal history of basins (including the post-rift phase), and sedimentation/erosion, all of which modulate the rheological stratification of the lithosphere. Experiments by Buiter et al. (2009) demonstrate that basin inversion is promoted mainly by: (1) mechanically weak basin fill (relative to the basement rocks), (2) strain-weakened, basin-bounding shear zones or normal faults, and (3) the erosion of sedimentary overburden once basin inversion begins, which facilitates isostatic uplift and further reduces the brittle strength of the crust. Such experiments show that during shortening, localised viscous deformation and isostasy contribute to strain localisation and uplift along pre-existing rift basins.

Lithospheric-scale analogue experiments can be a useful tool for investigating the influence of the lithosphere underneath rift basins, especially its mechanical stratification, in promoting or suppressing basin inversion (e.g., Gartrell et al., 2005; Cerca et al., 2010). Such models allow us to investigate how the interaction between brittle and viscous deformation drives inversion, from the scale of an individual basin to an entire system of basins. While isothermal analogue models do not specifically take into account the thermal structure and evolution of the studied system, model parameters can be chosen such that the experiments simulate first-order natural rift- and inversion-related processes (e.g., upwelling of mantle material under thinned lithosphere due to rifting).

In this paper, we introduce a series of isothermal, lithospheric-scale analogue experiments that simulate continental extension (before reaching the necking and break-up stages) followed by shortening. The aim of our study was to investigate how pre-existing rift-related structures (i.e., rift basins and basin-bounding normal faults) affect deformation during a later contractional tectonic phase. In this contribution, we evaluate the influence of pre-rift rheological layering and the imposed bulk extension and shortening strain rate on: (1) the distribution of rift basins during extension and (2) whether all of these basins or only some of them are inverted during shortening. In any given inverted basin, we can assume that uplift of sedimentary infill is driven by uplift of the underlying basement (Figure 1). In our experiments, we observed the impact of shortening on the

topography of the model surface. We refer to the normal fault-bounded, topographic lows that formed during extension as "basins". As we did not introduce sedimentary infill during and following extension, we assume that the model surface is analogous to the top of the basement of natural rift basins (i.e., pre-rift rocks). Therefore, we consider a basin to be inverted when the top surface of that basement is displaced upwards.

Our experimental setup is inspired by the Proterozoic basins of the North Australian Craton (northern Australia; Betts et al.,
2006), which have long drawn the interest of the petroleum and mineral exploration industries. The mineral-rich lithologies of these basins and their multistage history have been associated with the formation of world class mineral deposits, including the world's single largest source of sediment-hosted Pb–Zn deposits (Mount Isa, Queensland; Betts et al., 2003; Large et al., 2005; Gibson et al., 2016; Gibson and Edwards, 2020), the planet's oldest oil deposits (Northern Mount Isa Basin, Queensland; McConachie, 1993), and conventional and unconventional gas (Greater McArthur Basin, Northern Territory; Cox et al., 2022).
This distributed system of intra-cratonic basins in the North Australian Craton underwent multiple phases of extensional and compressional deformation driven by far-field plate boundary processes (Giles et al., 2002; Cawood and Korsch, 2008; Betts and Giles, 2006; Betts et al., 2008, 2011; Scott et al., 2000; Gibson et al., 2008). Our experiments are comparable to the initial basin-forming extensional phase (ca. 1800–1750 Ma; Jackson et al., 2000; Betts et al., 2006) and the shortening phase (ca. 1750–1710 Ma; Betts, 1999; Blaikie et al., 2017; Spence et al., 2021) that followed extension.

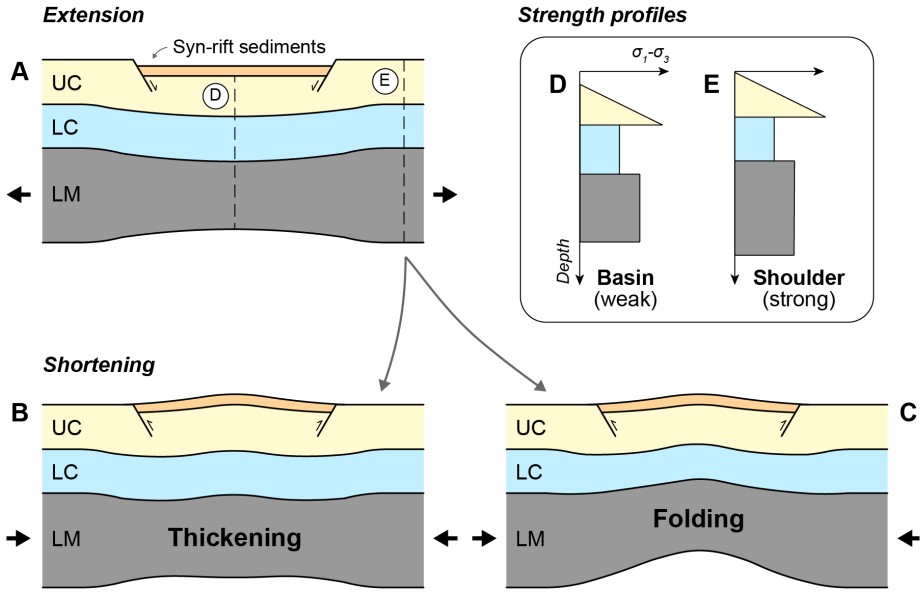


**Figure 1: Hypothesised deformation of a lithospheric-scale three-layer analogue model (supported by a liquid asthenosphere, not pictured) during extension and subsequent shortening. The model lithosphere comprises a brittle upper crust (UC), weak ductile lower crust (LC), and strong ductile lithospheric mantle (LM). During extension (a), localised thinning of the strong lithospheric mantle correlates with normal faulting and rift basin formation in the upper crust. During shortening, basin inversion could**
**potentially be driven by thickening (b) or folding and upwelling (c) of the ductile lower crust and lithospheric mantle (cf. Zwaan and Schreurs, 2023). This viscous deformation is accompanied by the reactivation of weakened, rift-related normal faults in a reverse sense (Marques and Nogueira, 2008; Buiter et al., 2009). (d and e) Comparison between strength profiles in the middle of a rift basin and at the rift shoulder.**

The experiments presented here highlight that the initial rheological layering of the models (which represents the thermal and compositional layering of the lithosphere) and the imposed kinematic boundary conditions (i.e., rate of rifting) influence rift evolution and the distribution and segmentation of rift basins. In turn, the distribution of these basins and the rheology of the model layers at the end of extension determines which of the basins are inverted during shortening. This selective uplift of basins in the (brittle) upper crust layer, which has not been observed in previous crustal and lithospheric-scale models of basin

inversion, appears to be controlled by viscous deformation of the (ductile) lower crust and lithospheric mantle layers.

## 2 Experimental method

The experimental setup for extension followed by shortening is illustrated in Figure 2. The initial objective of the experiments presented here was to identify a suitable reference experiment of wide rifting and subsequent shortening, against which future experiments (e.g., those that include pre-existing weaknesses) can be compared. Hence, multiple parameters were changed

between experiments (Table 1). Ultimately, we concluded that Models R4 and R5 were the most appropriate reference experiments for future experimental series on multistage tectonics in the North Australian Craton.

Models R1 and R2 consisted of an upper crust that was very thin relative to the ductile lower crust. In contrast, the thickness ratio between the upper and lower crust in Models R3, R4, and R5 was 50:50, which is more representative of the North Australian Craton (Section 2.1). The extension and shortening velocity for Models R1 and R2 was also much slower than for

R3, R4, and R5. As a result, the lithospheric mantle in R1 and R2 is weak compared to R3, R4, and R5 (Figure 2e), resulting in differences in the strain localisation behaviour of the lithospheric mantle (Section 3).

In Models R2 and R5, we introduced a cut (dipping 45°, striking orthogonal to the extension direction, and positioned approximately 13 cm from the "southern" boundary of the model) in the lithospheric mantle using a lubricated knife before the start of extension. This cut represented a "pre-rift" discrete weakness in the lithospheric mantle and was introduced to test

its influence on strain localisation during rifting (cf. Figure 1 in Santimano and Pysklywec, 2020). However, we found in the results (Section 3.2) that this pre-rift structure was not sufficient to localise extensional strain, and that the lithospheric mantle still behaved as an initially homogeneous layer. Therefore, the results of extension in Models R2 and R5 were comparable to Models R1 and R4, respectively. Further details on boundary conditions, initial conditions, and scaling are included in the following sub-sections.

**2.1 Boundary and initial conditions**

The model layers comprise a granular "upper crust", ductile "lower crust", and ductile "lithospheric mantle". The ductile materials exhibit spatially continuous deformation at the scale of observation. They behave viscously under our range of experimental strain rates, simulating deformation in the viscous layers of the lithosphere (i.e., the lower crust and lithospheric

mantle). The brittle-ductile layers are isostatically supported by a liquid that is analogous to the natural asthenosphere (Figure 2a).

The yield strength profiles of the models resemble natural lithospheric strength profiles. The model strength profiles include a relatively strong upper crust as well as lower crust and lithospheric mantle layers of varying relative strengths (Figure 2e; Table 1). As the experiments were designed to help us better understand Proterozoic craton-wide rifting in the North Australian Craton (Allen et al., 2015), we implemented a rheological layering that allowed extension to be relatively uniform across the entire model area and create a distributed system of basins (i.e., "wide rifting" *sensu* Buck, 1991; also see Brun, 1999 and Buck et al. 1999). Hence the model lithosphere is analogous to a natural thick lithosphere (with a thick crust) shortly after orogenesis or with a higher-than-normal heat flow (Buck et al., 1999). Given the challenge of reconstructing the lithosphere configuration and rifting conditions of the North Australian Craton in the Proterozoic, we used the Basin and Range Province – a well-known example of a wide rift (e.g., Hamilton, 1987; Parsons, 2006) – as a proxy for estimating crustal thicknesses (Gueydan et al., 2008) and the rate of extension for our models. Hence, the thicknesses of the crustal layers in Models R1 and R2 scale to 10 km and 40 km for the upper and lower crust, respectively. After running Models R1 and R2, we found that it would be more representative of the North Australian Craton (Betts et al., 2002; Kennett et al., 2011) to have upper and lower crust layers with the same thickness, which we then implemented in Models R3, R4, and R5 (Table 1).

The models were extended at a velocity that scales to 1–2 cm/year in nature (Table 2), which is within the range of estimated rates of extension for the Basin and Range Province (Bennett et al., 1998; Snow and Wernicke, 2000; Hammond and Thatcher, 2004; Tetreault and Buiter, 2018). After ~8 cm of extension (~20% bulk extension), the models were shortened in the reverse direction at the same rate until they reached their initial pre-extension length (Figure 2), simulating orthogonal rifting and then shortening. Before the start of extension, dark coffee powder was sifted onto the light-coloured model surface, which created high-contrast speckle patterns for deformation monitoring (Section 2.3). More coffee powder was sifted after the end of extension to ensure that there were sufficient dark particles inside the newly formed rift basins. As it took time to add coffee powder to the model surface, ensure that there were no problems with recording during extension, and initiate the shortening part of the experiment, seven to 24 minutes elapsed between the end of extension and the start of shortening. This pause scales to a maximum of ~1.7 Myr in nature (not considered significant compared to the entire duration of extension and shortening; see Section 2.2 for details on scaling factors). Model R1 is an exception, as no shortening was applied after extension.

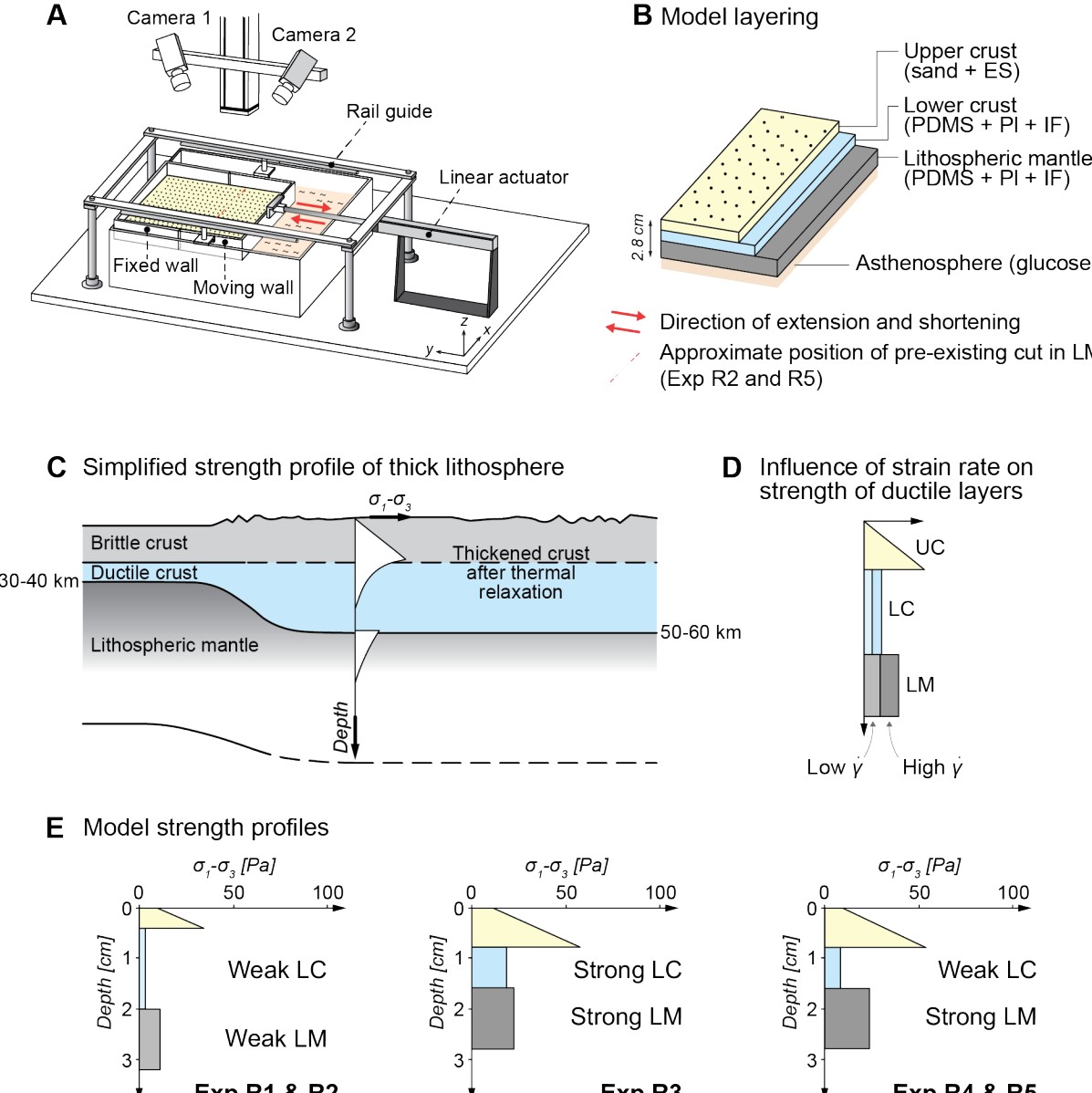

Figure 2: Experimental setup. The red arrows indicate the imposed extension and shortening directions. (b) Cross section of model layers. (a) Natural strength profile of a thickened lithosphere (including a thickened crust) after orogenesis and thermal relaxation, which forms widely distributed grabens upon extension (Brun, 1999). (c) Natural strength profile of a thickened lithosphere (including a thickened crust) after orogenesis and thermal relaxation, which forms widely distributed grabens upon extension (Brun, 1999). (d) Example of three-layer analogue model strength profile, showing that the strength of the ductile layer increases with strain rate $\dot{\gamma}$ (after Brun, 1999). (e) Initial strength profiles of models in this study. UC = upper crust; LC = lower crust; LM = lithospheric mantle; ES = Envirospheres; PDMS = polydimethylsiloxane; Pl = plasticine; IF = iron filings.

**Table 1 Summary of experimental parameters (UC = upper crust; LC = lower crust; LM = lithospheric mantle).**

| Exp | Layer thickness | | | Extension & shortening | | Brittle-ductile thickness ratio | Experimental strain rate | LC:LM strength ratio | Layer strength | |
|---|---|---|---|---|---|---|---|---|---|---|
| | UC | LC | LM | Velocity | Duration | | | | LC | LM |
| | [cm] | [cm] | [cm] | [mm h$^{-1}$] | [h] | | [s$^{-1}$] | | [Pa] | [Pa] |
| R1 | 0.4 | 1.6 | 1.2 | 6.2 | 14.0 | 0.1 | 5.4 x 10$^{-5}$ | 0.4 | 4.3 | 10.7 |
| R2 | 0.4 | 1.6 | 1.2 | 6.2 | 14.0 | 0.1 | 5.4 x 10$^{-5}$ | 0.4 | 4.3 | 10.7 |
| R3 | 0.8 | 0.8 | 1.2 | 31.0 | 3.0 | 0.4 | 3.1 x 10$^{-4}$ | 0.8 | 18.5 | 21.8 |
| R4 | 0.8 | 0.8 | 1.2 | 28.3 | 3.1 | 0.4 | 2.8 x 10$^{-4}$ | 0.2 | 11.23 | 68.51 |
| R5 | 0.8 | 0.8 | 1.2 | 28.3 | 3.1 | 0.4 | 2.8 x 10$^{-4}$ | 0.2 | 11.23 | 68.51 |

## 2.2 Scaling parameters and rheology of model materials

The model lithosphere layers were created using granular and ductile materials similar to those used by Molnar et al. (2017)
and Samsu et al. (2021). The brittle upper crust, the behaviour of which can be described using Mohr–Coulomb law (Byerlee, 1978), was modelled using a granular mixture comprising 89% dry quartz sand (Rocla 90 Fine Foundry Sand, Hanson Australia) and 11% hollow ceramic Envirospheres®. The sand is fine to medium-grained, with 78% of the grain sizes falling between 150 and 425 μm. The Envirospheres® were added to the sand to ensure that the density of the model upper crust is lower than that of the lower crust (Table 2) and still scales appropriately to the natural upper crust (c.f., 2670 kg/m$^3$; Artemjev
and Kaban, 1994). Using a Hubbert-type shear box, we measured an internal friction angle $\varphi \sim 49°$ and maximum extrapolated cohesion $C \sim 120$ Pa for the granular mixture. These values are high compared to a similar but slightly coarser-grained mixture used by Molnar et al. (2017) ($\varphi < 38°$ and cohesion $C \sim 9$ Pa; 75% of grains between 435 and 500 μm), the latter of which seemed more consistent with the ≤60° dip of the normal fault surfaces from the extensional phase of the experiments. Therefore, we adopted the internal friction angle and cohesion of the granular mixture used by Molnar et al. (2017) for the
calculation of differential stress in the brittle layer of the strength profiles.

The ductile lower crust and lithospheric mantle layers were modelled using mixtures that mostly consist of polydimethylsiloxane (PDMS). For the lithospheric mantle, black Colorific® plasticine was added to the PDMS in order to increase its effective viscosity (Boutelier et al., 2008). On its own, PDMS is a Newtonian fluid (Weijermars, 1986), meaning that its viscosity is strain rate-independent. However, the PDMS-based mixtures used to model the lithospheric mantle in our
experiments are slightly non-Newtonian, as they exhibit strain rate-softening behaviour (stress exponent n ~ 1.4) (Table 2).

**Table 2** Scaling parameters for all experiments. Abbreviations of modelling materials: ESPH = Envirospheres, BPL = black Plasticine, IF$_{uf}$ = ultrafine iron filings, IF$_f$ = fine iron filings.

| Models R1 & R2 | | Thickness | | Density | | Viscosity | | Stress | |
|---|---|---|---|---|---|---|---|---|---|
| | | Model | Nature | Model | Nature | Model | Nature | exponent | |
| | | (mm) | (km) | (kg/m³) | (kg/m³) | (Pa s) | (Pa s) | | Material |
| Upper crust | Brittle | 4 | 10 | 1222 | 2650 | - | - | | Sand+ESPH |
| Lower crust (LC1) | Ductile | 16 | 40 | 1245 | 2700 | $6.0 \times 10^4$ | $2.2 \times 10^{21}$ | 1 | PDMS+IF$_{uf}$ |
| Lithospheric mantle (LM1) | Ductile | 12 | 30 | 1338 | 2900 | $3.6 \times 10^5$ | $1.3 \times 10^{22}$ | 1.36 | PDMS+BPL+IF$_{uf}$ |
| Asthenosphere | Fluid | - | - | 1430 | 3100 | 520 | $1.9 \times 10^{19}$ | | Glucose |
| **Scaling factors** | | $L^* = 4.0 \times 10^{-7}$ | | $\rho^* = 4.6 \times 10^{-1}$ | | $\eta^* = 2.7 \times 10^{-17}$ | | | |
| | | $t^* = 1.5 \times 10^{-10}$ | | $g^* = 1$ | | 1 h in model ~ 0.8 Myr in nature | | | |
| | | $v^* = 2.7 \times 10^3$ | | $\sigma^* = 1.9 \times 10^{-7}$ | | 3.1 mm/h in model ~ 10 mm/yr in nature | | | |

| Model R3 | | Thickness | | Density | | Viscosity | | Stress | |
|---|---|---|---|---|---|---|---|---|---|
| | | Model | Nature | Model | Nature | Model | Nature | exponent | |
| | | (mm) | (km) | (kg/m³) | (kg/m³) | (Pa s) | (Pa s) | | Material |
| Upper crust | Brittle | 8 | 20 | 1245 | 2700 | - | - | | Sand+ESPH |
| Lower crust (LC1) | Ductile | 8 | 20 | 1315 | 2850 | $6.0 \times 10^4$ | $2.2 \times 10^{22}$ | 1 | PDMS+IF$_{uf}$ |
| Lithospheric mantle (LM1) | Ductile | 12 | 30 | 1384 | 3000 | $2.1 \times 10^5$ | $7.8 \times 10^{22}$ | 1.36 | PDMS+BPL+IF$_{uf}$ |
| Asthenosphere | Fluid | - | - | 1430 | 3100 | $5.2 \times 10^2$ | $1.9 \times 10^{20}$ | | Glucose |
| **Scaling factors** | | $L^* = 4.0 \times 10^{-7}$ | | $\rho^* = 4.6 \times 10^{-1}$ | | $\eta^* = 2.7 \times 10^{-18}$ | | | |
| | | $t^* = 1.5 \times 10^{-11}$ | | $g^* = 1$ | | 1 h in model ~ 7.8 Myr in nature | | | |
| | | $v^* = 2.7 \times 10^4$ | | $\sigma^* = 1.9 \times 10^{-7}$ | | 31 mm/h in model ~ 10 mm/yr in nature | | | |

| Models R4 & R5 | | Thickness | | Density | | Viscosity | | Stress | |
|---|---|---|---|---|---|---|---|---|---|
| | | Model | Nature | Model | Nature | Model | Nature | exponent | |
| | | (mm) | (km) | (kg/m³) | (kg/m³) | (Pa s) | (Pa s) | | Material |
| Upper crust | Brittle | 8 | 20 | 1136 | 2700 | - | - | | Sand+ESPH |
| Lower crust (LC2) | Ductile | 8 | 20 | 1199 | 2850 | $3.0 \times 10^4$ | $1.1 \times 10^{22}$ | 1 | PDMS+IF$_f$ |
| Lithospheric mantle (LM2) | Ductile | 12 | 30 | 1304 | 3100 | $2.7 \times 10^5$ | $9.0 \times 10^{22}$ | 1.37 | PDMS+BPL+IF$_f$ |
| Asthenosphere | Fluid | - | - | 1430 | 3400 | 520 | $1.9 \times 10^{20}$ | | Glucose |
| **Scaling factors** | | $L^* = 4.0 \times 10^{-7}$ | | $\rho^* = 4.2 \times 10^{-1}$ | | $\eta^* = 2.7 \times 10^{-18}$ | | | |
| | | $t^* = 1.6 \times 10^{-11}$ | | $g^* = 1$ | | 1 h in model ~ 7.1 Myr in nature | | | |
| | | $v^* = 2.5 \times 10^4$ | | $\sigma^* = 1.7 \times 10^{-7}$ | | 28 mm/h in model ~ 10 mm/yr in nature | | | |


Iron filings were also added to the lower crust and lithospheric mantle material to increase their densities. For Models R4 and R5, the addition of fine-grained iron filings (0.42–0.82 mm grain size, manufactured for Chem-Supply Australia) did not significantly affect the flow behaviour of the PDMS-based mixture during our experiments (LC2 and LM2 in Table 2). However, in Models R1, R2, and R3, the use of ultrafine-grained iron filings (manufactured for Mad About Science), which

has a powder-like consistency, had the unintended effect of doubling the viscosity of the PDMS-based mixture (compare LC1 and LC2 in Table 2). We opted to use fine-grained iron filings for Models R4 and R5 to mitigate this viscosity increase. Hence the yield strength profile for Model R3 contains a "strong" lower crust and high LC:LM strength ratio, while Models R4 and

R5 contain a lower crust that is significantly weaker than the lithospheric mantle (low LC:LM strength ratio; Figure 2e, Table 1); the latter is more consistent with theoretical strength profiles for a wide rift setting (Brun, 1999).

The scaling parameters (Ramberg, 1967) used in our experiments and the properties of the model layers are presented in Table 2. These parameters were chosen so that model deformation is consistent with natural processes but occurs over a time scale that is convenient for laboratory experiments. The length scaling factor $L^* = L_m/L_p = 4 \times 10^{-7}$ meant that 0.4 cm in the model scales to 10 km in nature, whereby the subscripts $m$ and $p$ denote the model and natural prototype, respectively. Therefore, the model surface area of 44 cm x 40 cm corresponds to 1100 km x 1000 km in nature, and the model thicknesses of 2.8–3.2 cm

represent lithospheric thicknesses of 70–80 km. The density scaling factor $\rho^* = \rho_m/\rho_p$ was set to 0.46 based on the density ratio between the model asthenosphere material (Queen Glucose Syrup; Schellart, 2011) and estimated asthenosphere densities between 3100 kg/m$^3$ and 3400 kg/m$^3$, consistent with previous lithospheric-scale analogue experiments (e.g., Molnar et al., 2017; Santimano and Pysklywec, 2020; Samsu et al., 2021) and reference asthenospheric densities used in geophysical models (e.g., 3250 kg/m$^3$ in Lamb et al., 2020). Similarly, the viscosity scaling factor $\eta^* = \eta_m/\eta_p$ was determined using the ratio

between the effective viscosity of the model asthenosphere (520 Pa s) and that of the natural asthenosphere (1.9 x 10$^{19}$ Pa s for Models R1 and R2; 1.9 x 10$^{20}$ Pa s for Models R3, R4, and R5). As the experiments were conducted under normal gravitational acceleration, the scaling factor for acceleration due to gravity $g^* = g_m/g_p = 1$, resulting in a stress scaling factor $\sigma^* = \rho^* \times g^* \times L^*$ between 1.7 x 10$^{-7}$ and 1.9 x 10$^{-7}$. The above scaling factors were used to calculate the time scaling factor $t^* = \eta^* / (\rho^* \times g^* \times L^*)$ and velocity scaling factor $v^* = L^* / t^*$.

For Models R1 and R2, we started out with a natural asthenosphere density $\rho_p = 3100$ kg/m$^3$ and viscosity $\eta_p = 1.9 \times 10^{19}$ (following Molnar et al., 2017) and an extension velocity that scaled to 2 mm/year, resulting in an extension duration of 14 hours. For Model R3, the objective was to explore the behaviour of the ductile layers when we extended the model by the same amount but at a faster rate. Therefore, the prototype viscosity was increased by one order of magnitude (to $_p = 1.9 \times 10^{20}$) to achieve an appropriate time scaling factor. This change in the time scaling factor enabled us to apply an extension rate that

still scaled to 2 mm/year in nature within a shorter (experimental) extension duration, i.e., around 3 hours. However, additional changes to the ductile materials were still necessary, as the strength contrast between the lower crust and lithospheric mantle (LM1 in Table 2) in Model R3 was too low to simulate natural lithosphere with a strong lithospheric mantle and relatively weak lower crust (Table 1, Figure 2c). Therefore, for Models R4 and R5, we created an improved lithospheric mantle mixture (LM2 in Table 2) with the desired viscosity $\eta_m = 2.7 \times 10^5$ Pa s (approximately ten times greater than the model lower crust),

resulting in a low LC:LM strength ratio. As this mixture had a density $\rho_m = 1384$ kg/m$^3$, the density scaling factor was changed to 0.42 (using $\rho_p = 3400$ kg/m$^3$ for the asthenosphere), otherwise the prototype lithospheric mantle and asthenosphere densities would have both equalled 3100 kg/m$^3$. This last change did not significantly impact the other scaling factors. The layer densities in Models R4 and R5 are the most consistent with those used in geophysical models on the density structure of the lithosphere, where the densities of the upper crust, lower crust, and lithospheric mantle are 2700 kg/m$^3$, 2940 kg/m$^3$, and

3350 kg/m$^3$ respectively (Kaban et al., 2014).

The difference in the experimental strain rates between Models R1 and R2 and Models R3, R4, and R5 is a consequence of the difference in the viscosity scaling factor $v^*$ and therefore the time scaling factor $t^*$. The time scaling factor for Models R1 and R2 ($1.5 \times 10^{-10}$) is one order of magnitude higher than for R3 ($1.5 \times 10^{-11}$) and R4 and R5 ($1.6 \times 10^{-11}$). The experimental strain rate for Models R1 and R2 ($\sim 5.4 \times 10^{-5}$ s$^{-1}$) is one order of magnitude lower than for Models R3 ($\sim 3.1 \times 10^{-4}$ s$^{-1}$) and R4 and R5 ($\sim 2.8 \times 10^{-4}$ s$^{-1}$). These strain rates were estimated by dividing the rate of extension (i.e., the velocity of the moving wall during extension) by the initial thickness of the model lithosphere (Benes and Scott, 1996). As the strength of the ductile layers increases with the applied strain rate (Ranalli, 1995; Brun, 1999), the lithospheric mantle in Models R1 and R2 is weak compared to the lithospheric mantle in R3, R4, and R5 (Figure 2d). For R1 and R2, the strength profile shows that the lower crust and lithospheric mantle are both weak compared to the upper crust, due to the slow strain rate applied to these experiments.

The scaling parameters used in Model R3 and Models R4 and R5 are relatively similar, with the main difference being the effective viscosity of the lower crust in R3 being twice greater than in R4 and R5 (Table 2). As a result, in Model R3 the lower crust is almost as strong as the lithospheric mantle (Figure 2). In R4 and R5, the lower crust is much weaker than the lithospheric mantle. The relative strength of the lower crust with respect to the overlying and underlying layers affects the mechanical coupling between the upper crust and lithospheric mantle and therefore the strain distribution in the upper crust, as discussed further in Section 3.1.

**2.3 Deformation monitoring and analysis**

Digital image correlation (DIC) was applied to sequential images of the model surface in order to monitor deformation in the cover layer during the experiment. This technique allowed us to observe the strain and topographic evolution of the models. Strain maps and orthorectified photographs of the model surface were used to track the formation of rift basins and inversion structures at different stages of the experiments.

The image acquisition and DIC workflow is similar to that outlined in Molnar et al. (2017) and Samsu et al. (2021). The DIC system comprises two cameras at oblique angles to the model surface (Figure 2a). Images were recorded at five-minute intervals over 14 hours for experiments R1 and R2 and at two-minute intervals over approximately three hours for experiments R3, R4, and R5 for each extension or shortening phase. Surface strain and topography were computed using the StrainMaster module of the commercial image correlation software DaVis (version 10.1.2, LaVision). The software uses stereo cross correlation to compute the incremental displacement field from which the strain tensor components derived. In our experiments, high-contrast speckle patterns created by coffee powder sifted on the model surface were used for image correlation.

For the strain maps, the displacement vector fields obtained from DaVis were used to derive incremental and cumulative axial strain ($e_{yy}$ and $E_{yy}$, respectively) in MATLAB. $e_{yy}$ and $E_{yy}$ are measures for normal strain parallel to the extension and shortening direction. $E_{yy}$ was computed from the displacement gradient tensor. The displacement gradient tensor $H$ comprises the

components $\frac{\Delta D_i}{\Delta x_j}$, where $D_i$ are the displacement components in the x- and y-direction (i.e., $D_u$ and $D_v$) and $x_i$ refer to the x- and y-axis of the coordinate system (with the y-axis being parallel to the extension and shortening axis). Using the Lagrangian finite strain tensor $E$ (Allmendinger et al., 2011):

$$E = \begin{bmatrix} E_{xx} & E_{xy} \\ E_{yx} & E_{yy} \end{bmatrix} \tag{1}$$

The normal strain along the extension and shortening axis can be calculated with:

$$E_{yy} = \frac{1}{2}\left( \frac{\Delta D_u}{\Delta y} + \frac{\Delta D_v}{\Delta y} + \left( \frac{\Delta D_u}{\Delta y}\frac{\Delta D_u}{\Delta y} + \frac{\Delta D_v}{\Delta y}\frac{\Delta D_v}{\Delta y} \right) \right) \tag{2}$$

The incremental vertical displacement ($d_w$), cumulative vertical displacement ($D_w$), and the height (i.e., topography) of the model surface were also calculated in DaVis.

The incremental and cumulative displacement field data generated by DaVis were imported into MATLAB for post-processing for visualisation purposes. Post-processing of data included detection and replacement of spurious displacement vectors and interpolation of missing data. To this end, we used the DCT-PLS algorithm (Garcia, 2011). Topographic data was corrected by fitting a plane through the initially flat but possibly tilted model surface. Finally, the resulting linear correction parameters were applied to all subsequent digital elevation models. The MATLAB scripts used for post-processing and visualisation are available at https://github.com/TimothySchmid/PIV_postprocessing_2.0.

## 3 Results

Here we present the results of five experiments titled R1 to R5 (Table 2). With the exception of R1, each experiment comprises an "extension" and subsequent "shortening" phase, discussed separately in the subsections below. The resulting fault strikes and basin long axes are roughly perpendicular to the extension and shortening direction, given the kinematic boundary conditions that simulate orthogonal rifting and shortening. When viewing the models in map view, the upper/top side of the image is referred to as "north", and the model is being extended towards the "south" (bottom of the image). Curvature of the deformation features near the western and eastern model edges results from friction between the model edges and the confining U-shaped walls. We therefore limit our analysis to the central area that is unaffected by these boundary effects.

## 3.1 Extension: Normal faulting and basin formation

In all of our experiments, the imposed bulk extension of the model resulted in an extension-orthogonal, E-W trending horst and graben system. Here we define "graben" as a topographic depression bounded by parallel normal faults (Reid et al., 1913; Peacock et al., 2000) which accommodate the bulk extension imposed on the models (Figure 3, Figure 4, Figure 6). We also use the terms "graben" and "basin" interchangeably. The wide distribution of basins is analogous to natural examples of wide rifting, such as in the Proterozoic North Australian Craton (Allen et al., 2015; Betts et al., 2008), Basin and Range Province (Wernicke et al., 1988), Aegean Sea (Doutsos and Kokkalas, 2001), and East China Rift System (Tian et al., 1992).

In the experiments, rift evolution occurred in two main phases: (1) rift basin formation associated with normal fault nucleation, growth, and linkage; and (2) basin deepening and widening. Basins became fully established when the normal basin-bounding faults reached their final length. Progressive extension allowed the throw along the faults to increase, resulting in deepening of the grabens. Downward fault propagation was limited by the thickness of the upper crust, after which strain was accommodated by widening of the grabens.

The timing of the above processes with respect to amount of bulk extension varied with different model setups (see initial strength profiles and boundary conditions in Figure 2e and Table 1). The rift system evolved most quickly in Models R4 and R5, followed by R3 and then R1 and R2. For example, the model topography shows that by 10% bulk extension, graben-bounding faults in Models R4 and R5 appear to have already reached their full lengths, while the faults in Models R1, R2, and R3 are still in the nucleation and growth stage (c.f., Figure 3, Figure 4).

The models also exhibit different degrees of strain distribution, in terms of the spacing of faults and grabens, at the end of the extensional phase (at 19–20% bulk extension; Figure 3, Figure 4). Grabens in Models R1 and R2 are evenly spaced across the model area. Their spacing is greater compared to Models R3, R4, and R5. They are highly segmented, bounded by normal faults with short and irregular (non-linear) fault traces. These grabens are also relatively shallow, due to the thin upper crust layer (Table 1).

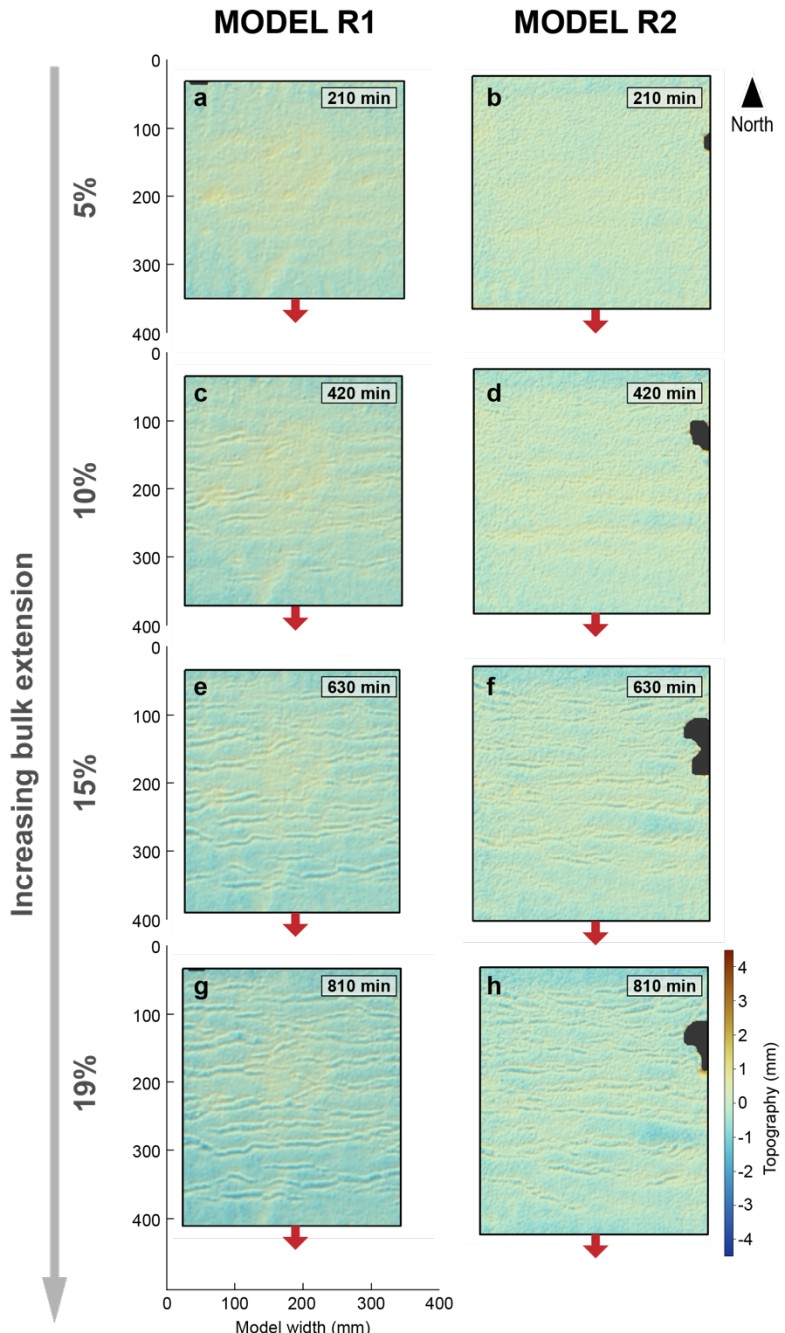

**Figure 3: Topography of Models R1 and R2 at increasing durations and amounts of extension applied to the model (e.g., a and b correspond to 5% bulk extension, c and d correspond to 10% bulk extension, etc). Arrows show the direction of extension. The resulting rift basins are uniformly spaced along the y-axis.**

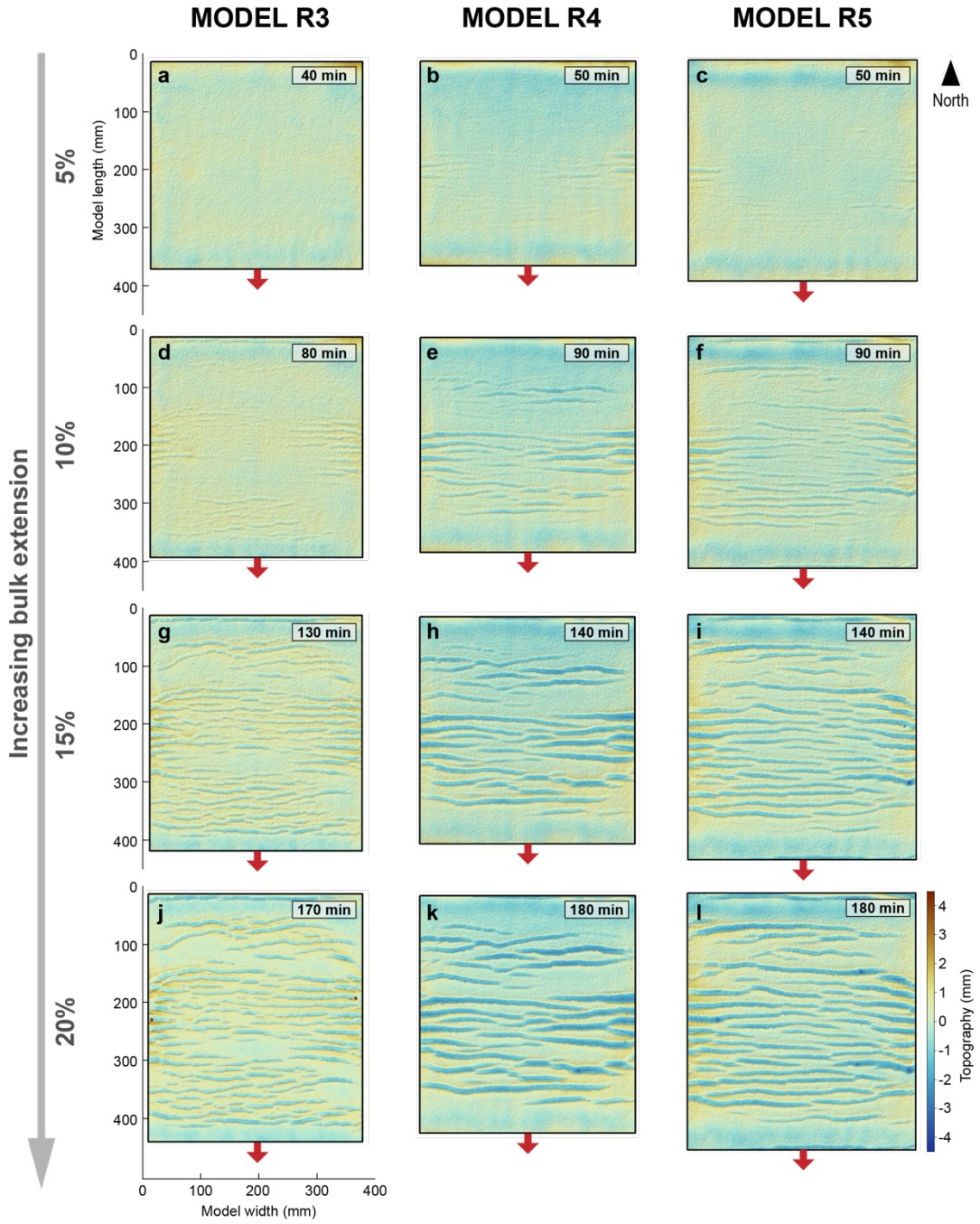

**Figure 4: Topography of Models R3, R4, and R5 at increasing durations and amounts of extension applied to the model (e.g., a, b, and c correspond to 5% bulk extension, c, d, and e correspond to 10% bulk extension, etc). The rift basins are not uniformly spaced. Arrows show the direction of extension.**

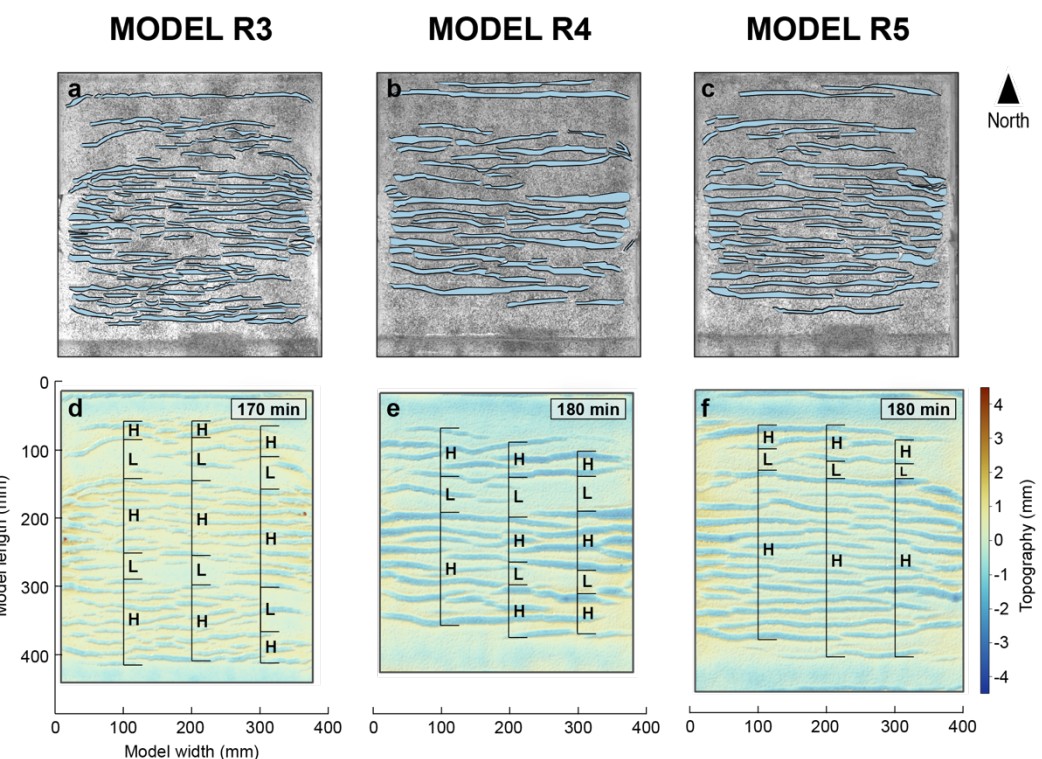

**Figure 5:** (a–c) Orthorectified top-view photos of the surface of Models R3, R4, and R5 at the end of extension, with overlay of the interpreted basins and basin-bounding fault traces. (d–f) High-strain (H) and low-strain (L) zones at the end of extension. Zones are marked along a profile at x = 100, 200, and 300 mm.

Fault traces in Models R3, R4, and R5 are more clearly defined than in Models R1 and R2, due to the lower quality of PIV data for Models R1 and R2 (possibly caused by sparser distribution of tracking particles on the model surface and non-optimal lighting conditions). Grabens in Model R3 are narrower and more segmented (i.e., less laterally continuous along the graben axes) than those in Models R4 and R5. Models R3, R4, and R5 exhibit clusters of grabens that make up so-called "high-strain zones", which are separated from each other by "low-strain zones" where the distance along the y-axis between two adjacent basins is greater than the widest basin in the model (Figure 5). Hence, basins are not distributed uniformly across the model area as they are in Models R1 and R2. There is no strain in the northernmost and southernmost ~5 cm of the model, as these segments were attached to the confining plexiglass walls; these areas are not included in the subsequent analyses. In the north of the model, there is a narrow low-strain zone adjacent to a wide high-strain zone, the latter of which makes up approximately the southern two-thirds of the model area. In Models R3 and R4, another low-strain zone is present within the wide high-strain zone (Figure 5).

 **3.2 Shortening**

### 3.2.1 Strain localisation at the boundaries of pre-existing basins

During shortening, strain is localised along pairs of basin-bounding normal faults that formed during the preceding extensional phase (Figure 6). At lower percentages of bulk shortening, $E_{yy}$ is localised along the edges of the grabens, suggesting that rift-related normal faults are reactivated in a reverse sense during shortening. In Models R3, R4, and R5, high strain accumulation during shortening occurs in the high-strain zones formed during the extensional phase. In addition, deformation is more intense in the southern part of the model compared to the north (see Model R5 example; Figure 6). The high cumulative strain in the south may be related to the increasing displacement gradient from north to south.

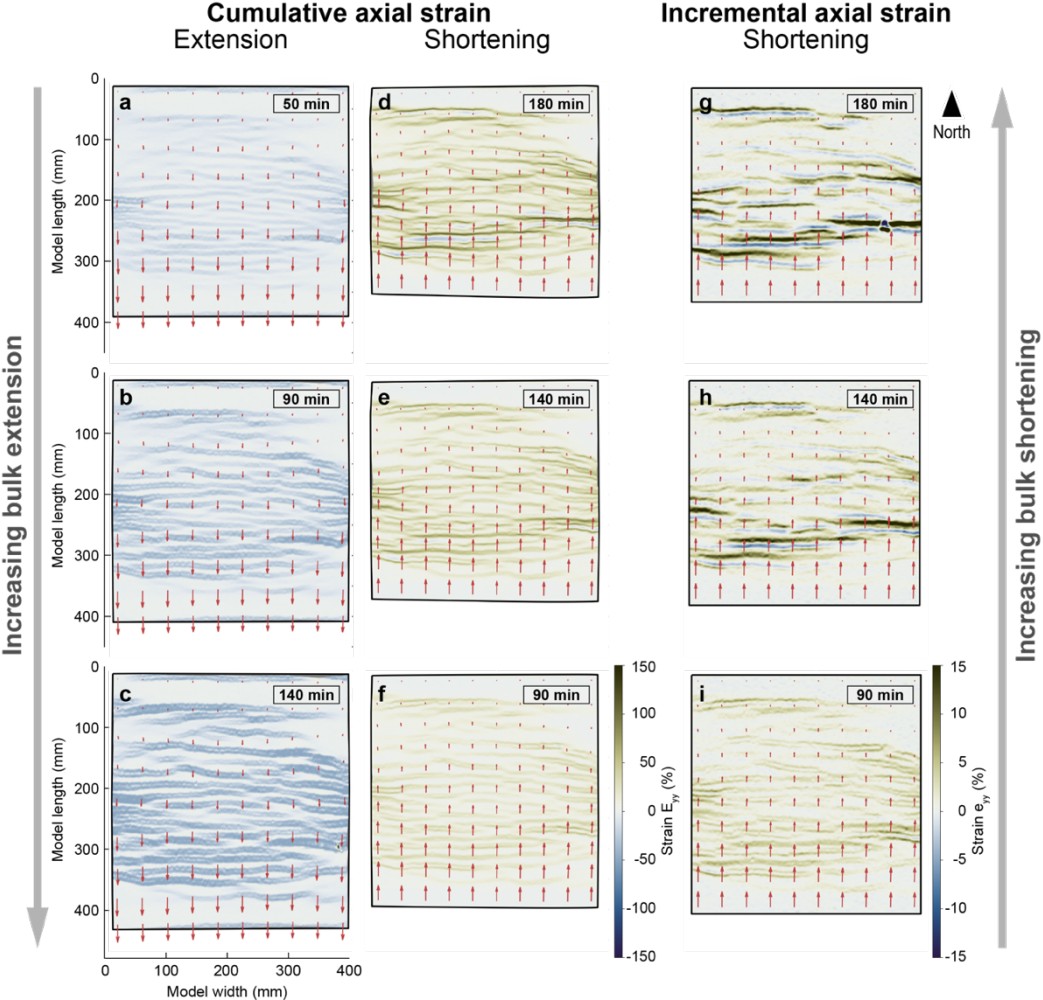

**Figure 6: Evolution of cumulative axial strain ($E_{yy}$) during extension and shortening (a-f) and incremental axial strain ($e_{yy}$) during shortening (g-i) in Model R5. During extension, $E_{yy}$ is localised along normal faults that form the edges of grabens (i.e., basin-bounding faults). During shortening, deformation is localised first along the same basin-bounding faults and then within the basins. Vector lengths represent relative amounts of displacement within the model.**

### 3.2.2 Correlation between axial strain and topography

High axial strain ($E_{yy}$) during shortening coincides with high vertical displacement ($D_w$) and high topography (Figure 7). We
interpret these linear, high-topography features to be analogous to inverted basins after low amounts of shortening and orogens
(i.e., mountain belts) after high amounts of shortening. The inverted basins and "orogens" on the surface of the model are
underlain by uplifted (ductile) lower crustal material, which can be observed after the granular upper crustal material has been
removed at the end of the experiments. A comparison of the topography of Models R3, R4, and R5 at the end of shortening
(~19–20% bulk shortening) shows that orogens are more laterally continuous (i.e., less segmented) when they form along
laterally continuous, pre-existing grabens (i.e., Models R4 and R5; Figure 8). In contrast, each elongate uplifted area in Model
R3 correlates with several segmented pre-existing basins.

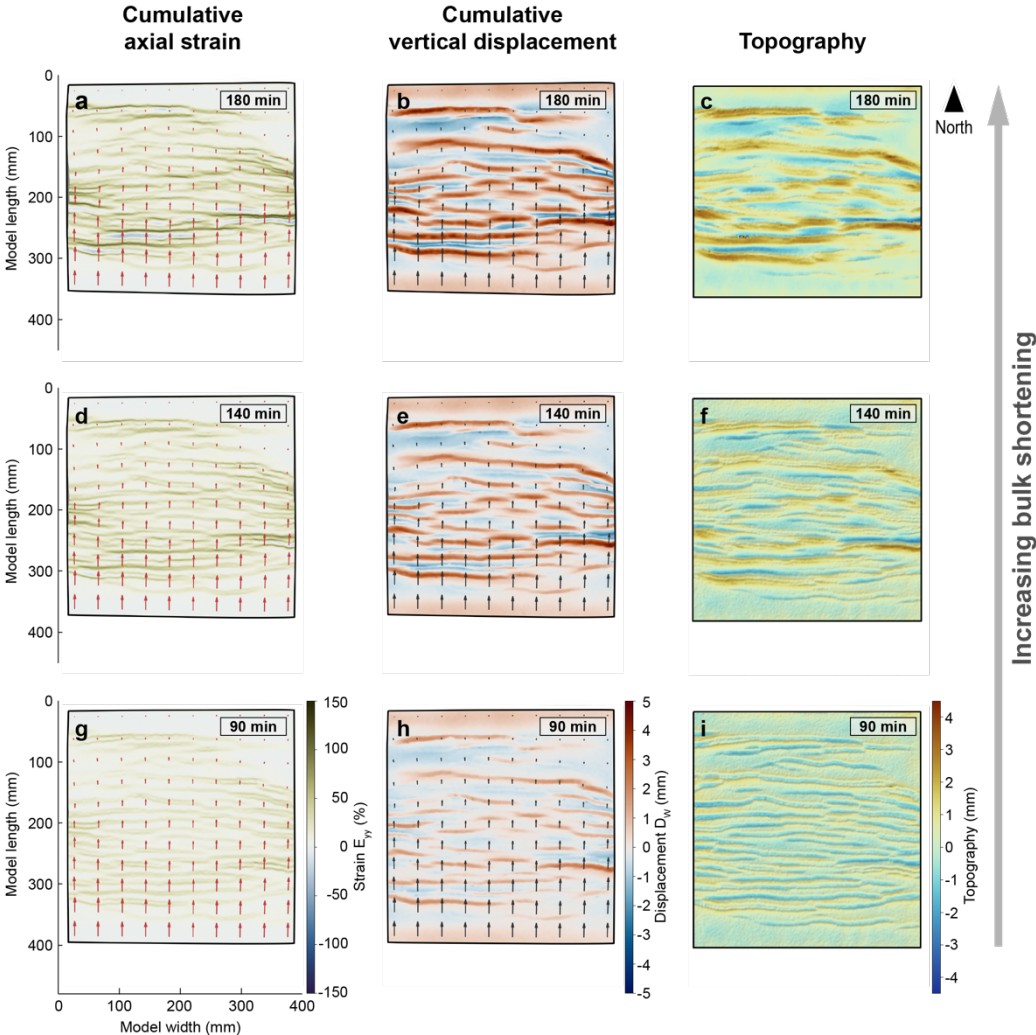

**Figure 7: Evolution of cumulative axial strain ($E_{yy}$), cumulative vertical displacement ($D_w$), and topography during shortening in Model R5.**

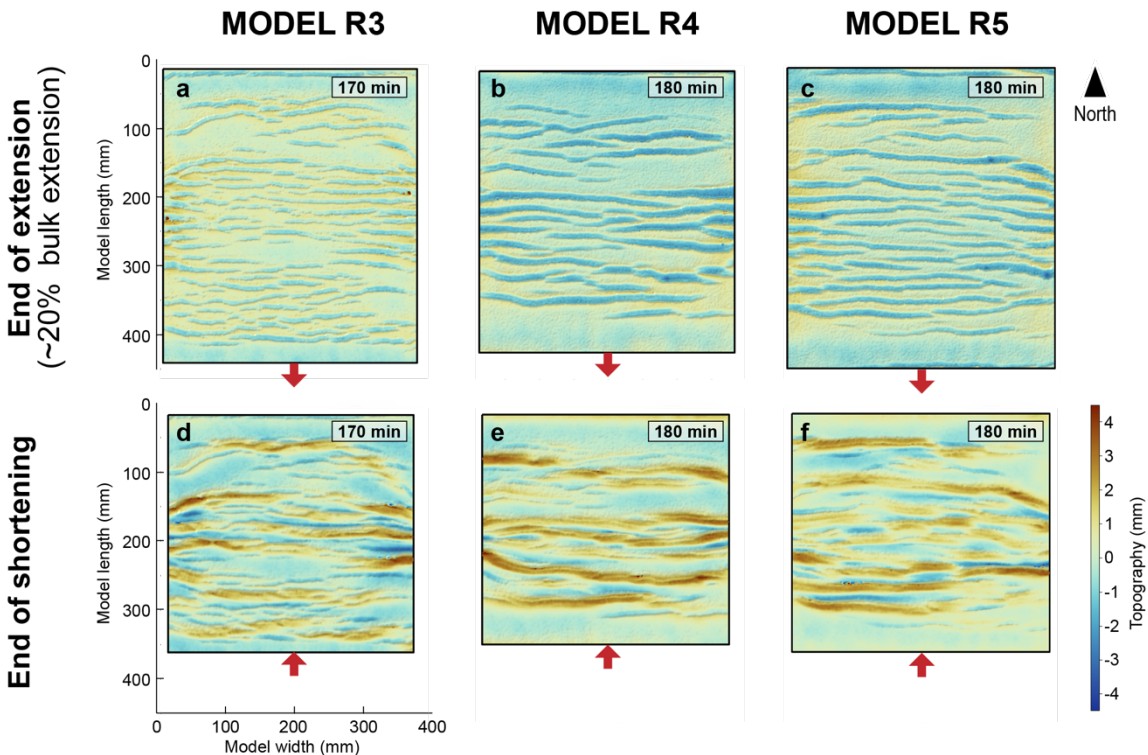

**Figure 8: Topography of models R3, R4, and R5 at the end of extension and shortening. Highly segmented extensional basins correlate with highly segmented orogens (R3). In contrast, laterally continuous extensional basins appear to localise laterally continuous orogens (R5).**

### 3.2.3 Selective uplift of basins

The topography of Models R3, R4, and R5 at the end of the extension and shortening phases (Figure 8 and Figure 9) suggests that some basins evolve into high-topography areas during shortening, while others remain as topographic lows. This selective inversion of basins is emphasised in the topographic profiles of Models R3 and R5 (Figure 10, Table 3 and Table 4). There appears to be a periodicity of uplift along the y-axis (or N-S axis) of the models, with regular spacing between basins that were eventually uplifted. The basins that remained as basins during shortening localised a high amount of axial strain in the direction opposite to shortening (negative axial strain in Figure 6). In contrast to Models R3, R4, and R5, all of the basins in Model R2 were uplifted during shortening. We discuss possible explanations for these uplift patterns in Section 4.2.

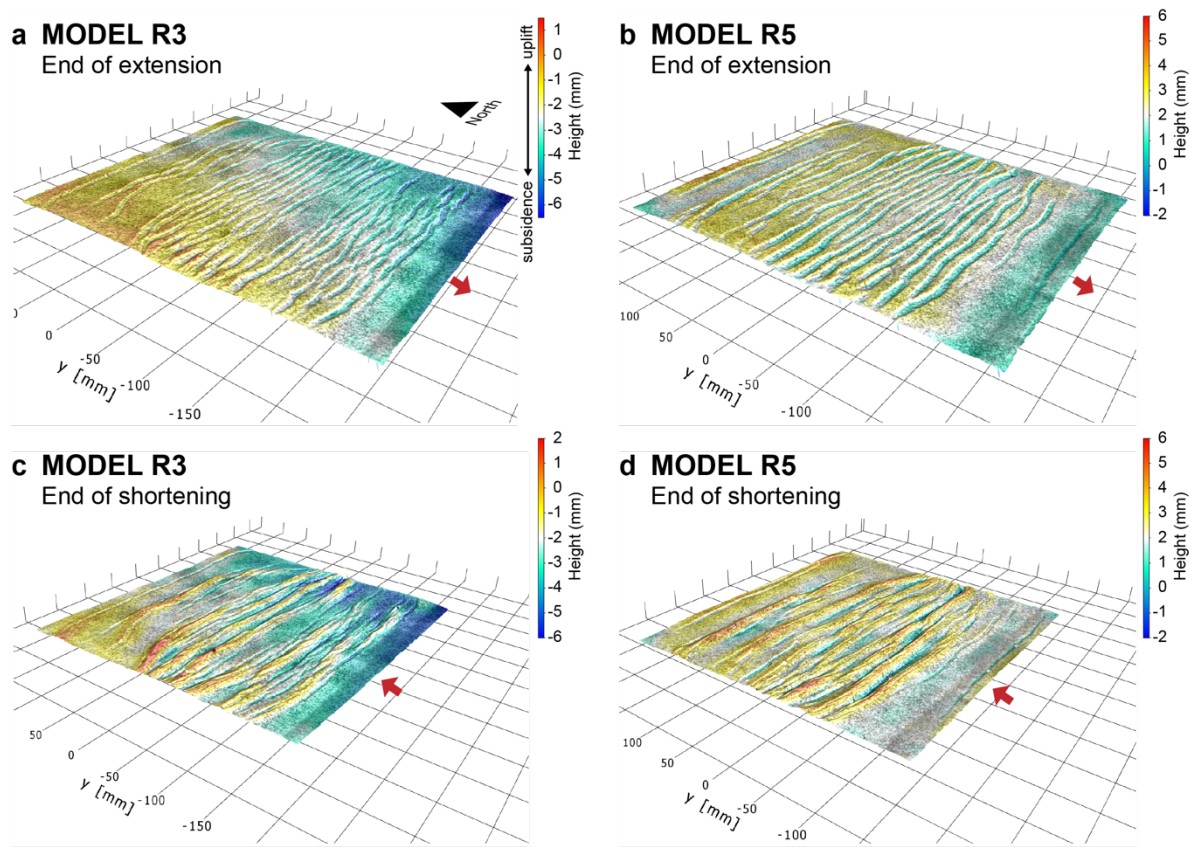

**Figure 9: Oblique 3D view of topography of Models R3 and R5 at the end of the extension and shortening phases. This 3D visualisation was done in DaVis prior to the postprocessing steps outlined in Section 2.3.**

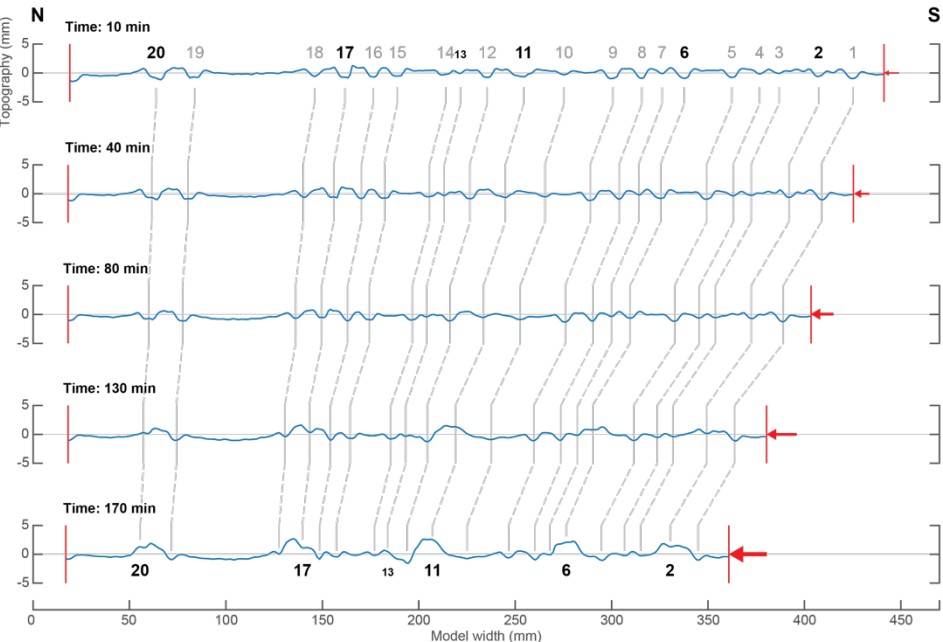

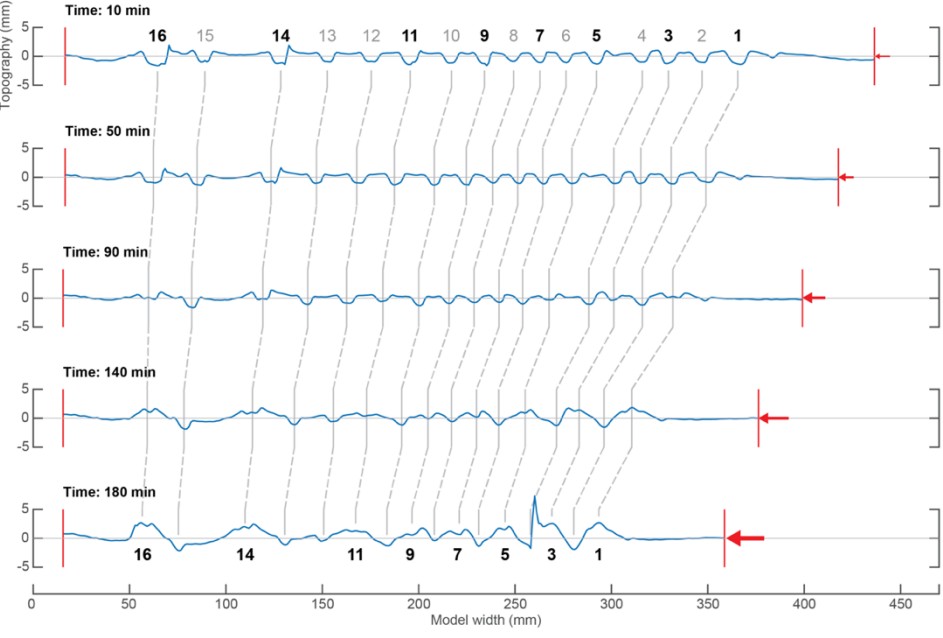

**Figure 10: Evolution of the N-S topographic profiles of Models R3 and R5 during shortening, drawn along x = 100 mm (see location in Figure 4). The numbers denote basins which had formed by the end of the extensional phase; numbers in bold correspond to basins that were uplifted during shortening. The red arrows represent the direction and cumulative amount of bulk shortening.**

**Table 3: Model R3 basin depths and positions 10 min after the start of shortening. Basins 2, 6, 11, 17, and 20 were uplifted by the end of shortening (Figure 10a); the even spacing between them suggests that the dominant wavelength for basin uplift (measured from the start of shortening) is between 69.2 mm and 98.1 mm. The increasing spacing between basins towards the north correlates with the lower displacement velocity towards the northern edge of the model.**

| Basin | y-position [mm] | Depth [mm] | Δy from previous basin [mm] | Δy from previous inverted basin [mm] |
|---|---|---|---|---|
| 1 | 424.6 | -1.0 | | |
| 2 | 406.1 | -0.6 | 18.5 | |
| 3 | 386.5 | -0.1 | 19.6 | |
| 4 | 376.1 | -0.2 | 10.4 | |
| 5 | 362.3 | -0.7 | 13.8 | |
| 6 | 336.9 | -0.9 | 25.4 | 69.2 |
| 7 | 325.3 | -0.1 | 11.5 | |
| 8 | 315.0 | -0.9 | 10.4 | |
| 9 | 300.0 | -0.9 | 15.0 | |
| 10 | 274.6 | -0.3 | 25.4 | |
| 11 | 252.7 | -0.6 | 21.9 | 84.2 |
| 12 | 234.2 | -0.8 | 18.5 | |
| 13 | 220.4 | -0.1 | 13.8 | |
| 14 | 212.3 | -0.3 | 8.1 | |
| 15 | 188.1 | -0.5 | 24.2 | |
| 16 | 175.4 | -0.7 | 12.7 | |
| 17 | 161.5 | -0.7 | 13.8 | 91.1 |
| 18 | 145.4 | -0.6 | 16.2 | |
| 19 | 83.1 | -0.7 | 62.3 | |
| 20 | 63.5 | -0.9 | 19.6 | 98.1 |

**Table 4: Model R5 basin depths and positions 10 min after the start of shortening. Basins 1, 3, 5, 7, 9, and 11 were uplifted by the end of shortening (Figure 10b); the even spacing between them suggests that the dominant wavelength for basin uplift (measured from the start of shortening) is between 29.0 mm and 38.4 mm. The greater distance between 11, 14, and 16 correlates with the lower displacement velocity towards the northern edge of the model.**

| Basin | y-position [mm] | Depth [mm] | Δy from previous basin [mm] | Δy from previous inverted basin [mm] |
|---|---|---|---|---|
| 1 | 366.9 | -1.4 | | |
| 2 | 347.2 | -1.1 | 19.7 | |
| 3 | 329.6 | -1.2 | 17.6 | 37.3 |
| 4 | 316.2 | -1.0 | 13.5 | |
| 5 | 292.3 | -1.3 | 23.8 | 37.3 |
| 6 | 276.8 | -1.1 | 15.5 | |
| 7 | 263.3 | -1.1 | 13.5 | 29.0 |
| 8 | 248.8 | -0.9 | 14.5 | |
| 9 | 234.3 | -1.2 | 14.5 | 29.0 |
| 10 | 216.6 | -1.1 | 17.6 | |
| 11 | 195.9 | -1.4 | 20.7 | 38.4 |
| 12 | 175.2 | -0.9 | 20.7 | |
| 13 | 153.4 | -1.0 | 21.8 | |
| 14 | 128.5 | -1.1 | 24.9 | 67.4 |
| 15 | 89.1 | -0.7 | 39.4 | |
| 16 | 64.3 | -1.6 | 24.9 | 64.3 |

## 4 Discussion

### 4.1 Rheological controls on rift basin distribution

Deformation of the extended lithosphere is accommodated by brittle faulting in the upper crust and viscous flow of the lower crust and lithospheric mantle. In our experiments, an initial period of distributed extension was followed by the localisation of deformation onto rift-related normal faults, which controlled the formation of rift basins (Figure 3, Figure 4, and Figure 6). The wide distribution of basins is consistent with previous extensional experiments of brittle-ductile models in which a rift seed was not implemented (e.g., Benes and Davy, 1996; Gartrell, 1997; Corti, 2005), which would have otherwise localised rifting from the onset of extension (i.e., "narrow rifting" in Buck, 1991). While all our models demonstrate wide rifting, different degrees of mechanical coupling between the model layers appear to have influenced the details of rift evolution (i.e., timing of basin formation) and the overall distribution of faults and basins.

In a wide rift mode, regular spacing between basins reflects the characteristic wavelengths of periodic instabilities during extension. These instabilities require a strength or viscosity contrast between two or more layers, and they form uniformly spaced domains of greater thinning, known as boudinage or pinch-and-swell structures (Ramberg, 1955; Smith, 1977). The formation of rift basins is controlled by two different wavelengths of periodic instabilities, which occur at a smaller scale in the brittle upper crust (i.e., crustal boudinage) and at the whole-lithosphere scale (i.e., lithospheric boudinage) (e.g., Benes and Davy, 1996). It is possible that the characteristic wavelengths of deformation localisation in our experiments is a product of the superposition of crustal and lithospheric boudinage, given their brittle-ductile, multi-layer setup. In Models R3 and R4, zones of localised lithospheric necking may correspond with high-strain zones, while areas that underwent minimal stretching may correspond with low-strain zones (Figure 4 and Figure 5). Here, the distances between the centres of high-strain zones may represent the characteristic wavelength of lithospheric-scale boudinage. In Model R4, the spacing between high strain zones is approximately 110 mm. The initial thickness of the ductile layers is 20 mm, giving a dominant wavelength/thickness ratio of 5.5, which is higher than the analytically predicted ratio of ~4 for lithospheric necking (Smith, 1977; Fletcher and Hallet, 1983). This disagreement may partly be due to the low stress exponent of our ductile layers ($n \leq 1.4$) and the influence of the overlying upper crustal sand layer. In Models R3, R4, and R5, basin spacing is on the order of the initial crustal thickness (18 mm). Benes and Davy (1996) observed a similar relationship between basin spacing and crustal thickness in their wide rift analogue experiments, from which they interpreted that the characteristic wavelength of crustal-scale periodic instabilities is on the order of the crustal thickness.

The coupling between the layers in analogue experiments is controlled by the relative strengths (i.e., effective viscosities) of the ductile layers, which is in turn influenced by the rate of extension (Zwaan et al., 2021; Brun, 1999). The rate of extension in Models R1 and R2 was much slower than in the other models (Table 1), so that the lithospheric mantle was relatively weak and underwent uniform thinning during extension. The thick and weak lower crust in R1 and R2 had sufficient time to flow during extension. As a result, both ductile layers thinned over a wide region, so that strain was distributed evenly in the overlying upper crust, resulting in evenly spaced basins from north to south (cf. Benes and Davy, 1996). The wide spacing

between the basins could be attributed to the large ratio of upper to lower crustal strength (Figure 2e), which predicts deformation localisation on a few structures (Brun, 1999) and large spacing between basins, with each basin-bounding fault

taking up a relatively large amount of strain (Wijns et al., 2005; Corti, 2005).

The faster rate of extension in Models R3, R4, and R5 resulted in a relatively strong lithospheric mantle compared to R1 and R2 (Figure 2e; cf. Brun, 1999; Nestola et al., 2015). This strong lithospheric mantle is overlain by a strong ductile lower crust in R3 and weak ductile lower crust in R4 and R5. Therefore, coupling between the lithospheric mantle and brittle upper crust is stronger in Model R3 than in Models R4 and R5. In previous experiments by Gartrell (1997), the brittle upper crust was

415 underlain by a strong, high-viscosity ductile layer (i.e., a so-called stress guide) and weak ductile lower crust. In their experiments, necking instabilities developed in the strong ductile layer and localised deformation into rift basins in the directly overlying upper crust. Similarly, our Model R3 consists of a strong ductile lower crust that directly underlies the upper crust and is almost as strong as the lithospheric mantle. Hence, the tight spacing between basins in Model R3 may correspond to short-wavelength localisation instabilities in the strong lower crust and lithospheric mantle. In Models R4 and R5, the weak

lower crust acted as a decoupling layer between the strong upper crust and strong lithospheric mantle. While this decoupling by an intervening weak layer does not appear to significantly influence the spacing between basins, it may have contributed to the formation of more laterally continuous basins of Models R4 and R5, as opposed to the short and segmented basins of Model R3 (cf. Benes and Davy, 1996). In any case, all of the experiments presented here demonstrate wide rifting, as predicted by Brun (1999) for a dominantly ductile lithosphere (brittle-ductile thickness ratio between 0.1 and 0.4; Table 1).

**4.2 Strain accommodation and basin inversion during shortening**

The evolution of axial strain in our models show that pre-existing rift basins exert a strong control on deformation related to far-field shortening. As normal basin-bounding faults formed in the upper crust during extension, they became zones of dilation within an otherwise undisturbed granular layer (Bellahsen and Daniel, 2005; Sassi et al., 1993; Mandl et al., 1977); these became pre-existing zones of weakness that were reactivated in a reverse sense in the early stages of shortening (Figure 6 and

430 Figure 7). The reverse reactivation of weakened normal faults during basin inversion has also been observed in previous analogue (e.g., Marques and Nogueira, 2008) and numerical experiments (e.g., Buiter et al., 2009). Steeply dipping normal faults with a ~60° dip are normally considered non-optimally oriented for dip-slip reactivation during subsequent shortening (e.g., Koopman et al., 1987; Brun and Nalpas, 1996; Eisenstadt and Sims, 2005). However, Bonini et al. (2012) suggested that reverse, dip-slip reactivation is indeed possible if a steeply dipping, pre-existing normal fault is rotated to shallower dips during

shortening and/or the principal stress axes are rotated. Stress rotation may be caused by shearing at the base of the brittle crust, so that the maximum principal stress axis deviates from the horizontal, and its angle with the pre-existing fault plane is reduced. It is possible that in our models, reactivation in the brittle crust is facilitated by the shallowing of the normal fault dips due to the upwelling of the underlying viscous material. In addition, shearing at the interface between the brittle and ductile layers and the presence of lateral heterogeneities after the extensional phase may have contributed to localised rotation of the principal

stress axes.

As shortening progressed, basins became narrower (Figure 7). These basins correspond with areas of previously thinned lithosphere (Figure 1), which would have been weaker than the rift shoulders. Continued shortening resulted in inversion of the basins, which we interpret to have been driven by anticlinal folding of the ductile layers, based on observations of uplifted lower crust underneath the inverted basins (following the removal of upper crustal material at the end of the experiments). This interpretation is comparable with observations from analogue experiments of continental collision (Sokoutis and Willingshofer, 2011) and intraplate compression (Dombrádi et al., 2010), where strain is accommodated and topography is controlled by folding of pre-existing weak zones. We also interpret that the anticlinal folding was facilitated by upward buoyancy forces where the upper crust (and therefore the lithosphere) was thinnest, in order to achieve isostatic equilibrium (Figure 11). This upwelling of viscous material underneath thinned crust or lithosphere has been observed in previous analogue models of rifting (Allemand and Brun, 1990; Brun and Beslier, 1996; Nestola et al., 2015; Beniest et al., 2018; Zwaan and Schreurs, 2023).

In their thermo-mechanical experiments of basin inversion, Sandiford et al. (2006) found that inversion is localised in the centre of the basin due to higher than average thermal gradients beneath the basin centre. In our analogue models, we observed that inversion is also greatest in the basin centre, based on maps of cumulative vertical displacement (Figure 7). Even though we cannot directly observe the influence of heat flow in our isothermal experiments, we speculate that at the end of extension, the basin centres in our models lie above thinned parts of the model lithosphere. In nature, lithospheric necking corresponds to a reduction in rift strength as hotter asthenospheric mantle material replaces colder and stronger lithospheric mantle (Chenin et al., 2018). In our models, lithospheric necking (Section 4.1) allows the weaker model asthenosphere to replace stronger lithosphere due to isostasy. Combining the results of Sandiford et al. (2006) and our experiments, we suggest that basin inversion in nature occurs where the lithosphere is weakest, and this zone of weakness is created by thinning of the strongest layers of the lithosphere and the upwelling of hot asthenospheric mantle material.

**4.3 Selective basin inversion due to periodic instabilities in the lithosphere**

The interpretation of thinned lithosphere facilitating inversion may be inadequate for explaining why only some basins were inverted in Models R3, R4, and R5 while others remained as basins (Figure 8, Figure 9, and Figure 10). In contrast, all of the pre-existing basins in Model R2 were inverted. Here we discuss factors that may have influenced: (1) whether all or only some rift basins are inverted during subsequent shortening (i.e., comparing Model R2 with Models R3, R4, and R5), and (2) the periodicity of selective basin inversion (i.e., comparing Model R3 with Models R4 and R5).

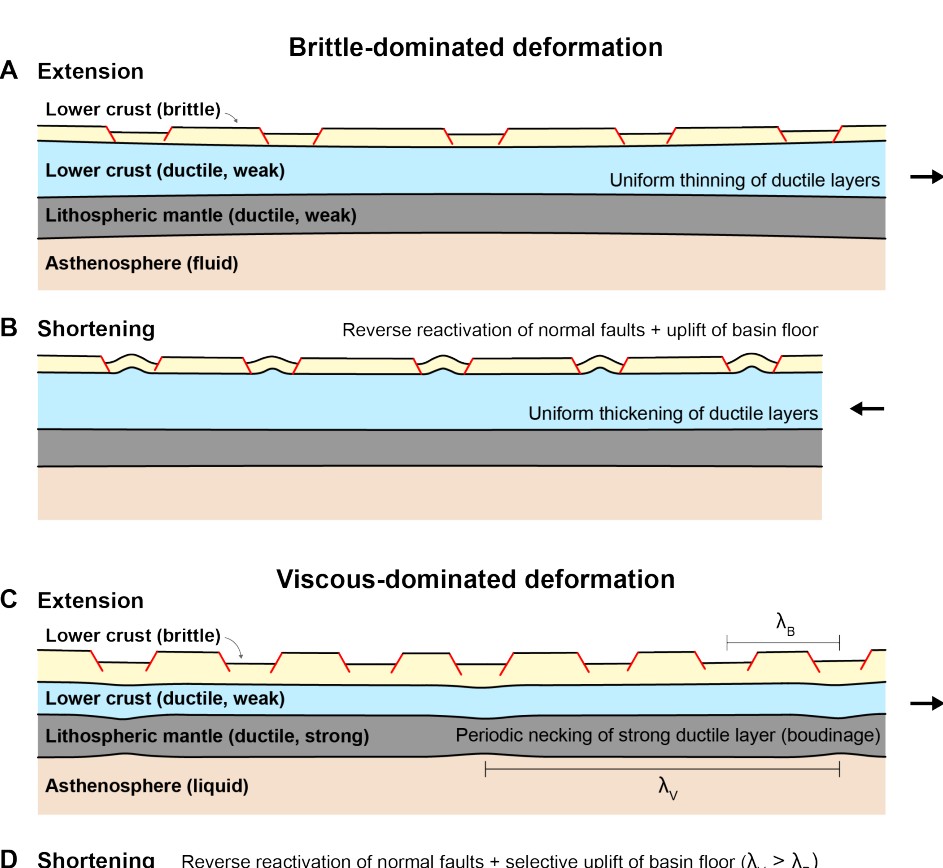

**Figure 11: Conceptual illustration of brittle-dominated deformation (e.g., R2) and viscous-dominated deformation (e.g., R3, R4, and R5) during post-extension shortening. Upper crustal deformation is observed directly from photographs and PIV-derived topographic and strain maps. Deformation of the ductile and liquid layers are inferred from observations of the top of the lower crust after the upper crust has been removed and are also inspired by previous analogue experiments (Benes and Davy, 1996; Gartrell, 1997). Brittle-dominated deformation occurs at slow strain rates, as deformation localisation is controlled by the frictional properties of the brittle upper crust (a and b). Viscous-dominated deformation corresponds with faster strain rates: During extension (c), lithospheric-scale boudinage occurs due to periodic instabilities (with a characteristic wavelength $\lambda_V$) in the ductile lithosphere layers. $\lambda_B$ denotes characteristic wavelength of strain localisation in the upper crust (i.e., the spacing between basins). When the lithosphere is shortened (d), the previously thinned ductile layers undergo folding, and the basins above these areas are inverted. Folding of the ductile layers, with the characteristic wavelength $\lambda_V$, may also occur even without previous boudinage (see explanation in Section 4.3).**

Model R2 was extended and shortened at a rate that was five times slower than Models R3, R4, and R5 (Table 1). Therefore, the ductile lower crust and lithospheric mantle layers of Model R2 were much weaker than the brittle upper crust. This meant that the ductile layers would have thinned and thickened uniformly during extension and subsequent shortening, respectively.

With no strain localisation in the ductile layers, rift-related faulting and basin formation in the granular, brittle upper crust during extension would have been controlled by the localisation of brittle deformation (Figure 10b and Figure 11a). During shortening of Model R2, upper crustal deformation may have been driven by the reverse reactivation of basin-bounding faults. Here there may have also been the significant influence of upwelling of ductile material underneath thinned crust, driving basin inversion. Despite the thinness (~4 mm) of the brittle upper crust in Model R2, the basins subsided enough to allow a locally thinned crust to form, creating a crustal weakness that is exploited for inversion.

The periodicity of basin inversion is only apparent when the models were extended and shortened at a sufficiently fast rate (i.e., Models R3, R4, and R5), which we interpret to have promoted localised viscous deformation as opposed to uniform thinning and thickening (i.e., Model R2). The inverted basins in Models R3, R4, and R5 may be underlain by uplifted (presumably folded) ductile lower crust. We assume that these anticlinal folds which were spaced evenly apart (Figure 10) represent periodic instabilities with a characteristic wavelength $\lambda_V$ (i.e., the distance between two anticlines; Figure 11c). Hence $\lambda_V$ also corresponds to the distance between two inverted basins. Based on our models, we conclude that for a system of distributed basins, where the distance between basins ($\lambda_B$) is shorter than $\lambda_V$, only some basins will be inverted (Figure 11d). Models of biharmonic folding (i.e., decoupled folding of the upper crust and lithospheric mantle, with two different wavelengths; Cloetingh et al., 1999) support the idea that topography can be controlled by the superposition of strain localisation at different levels of the lithosphere, with different characteristic wavelengths.

The wavelength and amplitude of folds during layer-parallel shortening is controlled by the thickness and rheology of the folded layers (e.g., Schmalholz and Mancktelow, 2016). While it is outside the scope of this work to analyse in detail how $\lambda_V$ is influenced by the model setup (e.g., layer thicknesses, viscosity ratios, bulk strain rates), we introduce here some simple calculations to assess whether folding of the ductile lithosphere layers is plausible. By treating the combined ductile lower crust and lithospheric mantle as a single ductile layer resting on a homogeneous viscous medium, we estimate an Argand number, $Ar \approx 3$ for Models R3, R4, and R5, using (Schmalholz et al., 2002):

$$Ar = \frac{\Delta \rho \, g \, H}{2 \, \mu_{eff} \, \dot{\varepsilon}_B} \qquad (3),$$

where $\Delta\rho$ is the density difference between the liquid asthenosphere and air, $g$ is the gravitational acceleration, $H$ is the combined thickness of the lower crust and lithospheric mantle, $\mu_{eff}$ is the average effective viscosity of the lower crust and lithospheric mantle, and $\dot{\varepsilon}_B$ is the experimental strain rate. An $Ar$ of this magnitude suggests that any folding that would occur in our models would be controlled by gravity. In the gravity-controlled mode of folding, the maximal amplification rate (i.e., growth rate) of folds $\alpha_{grav}$ can be calculated from (Schmalholz et al., 2002):

$$\alpha_{grav} = \frac{6n}{Ar} \, \dot{\varepsilon}_B \qquad (4),$$

with $n$ being the power law exponent of the ductile layer. In Models R3, R4, and R5, the calculated growth rate varies between 0.0006 and 0.0007, which is approximately 2 times greater than the experimental strain rate. Hence it is theoretically possible for gravity-controlled folding to have occurred in the experiments (Schmalholz et al., 2002) (Figure 11d). In contrast, folding

was theoretically insignificant in Model R2, which is consistent with a calculated $Ar \sim 17$ and a maximum amplification rate of $2.2 \times 10^{-5}$ (approximately 2 times less than the experimental strain rate). For the latter case, shortening was most likely accommodated by homogeneous layer thickening (Figure 11b) or inverse boudinage (Zuber, 1987).

The selective inversion of rift basins in our models has not been observed in previous crustal- and lithospheric-scale analogue experiments. There are few other analogue experiments in which extension followed by shortening is applied to a brittle-ductile model during the same experimental run. Examples of such experiments include the work of Gartrell et al. (2005) and Cerca et al. (2010), where extension is followed by shortening in a direction that is oblique to the extension direction (by 10° and 15°, respectively). However, these experiments differed from ours in that extension was localised by initial zones of weakness. As a result, the rift basins were not distributed across the entire model area, and all of the extensional basins localised subsequent shortening and associated inversion structures.

Our experiments show the importance of conducting lithospheric-scale analogue experiments – with a brittle-ductile multi-layer model underlain by a liquid asthenosphere for isostatic support – to investigate rheological controls on basin inversion. Future investigations on selective basin inversion would need to take into account sedimentation, as the density of basin sediments may suppress folding and uplift of the ductile layers during shortening. The lack of model cross sections in this investigation, which are challenging to make for a three-layer model lithosphere resting on glucose, hinders direct observations of important processes that are invoked, such as lithospheric boudinage and the upwelling of ductile and liquid material. Imaging of the ductile layers during experimental runs (e.g., using a CT-scanner; Colletta et al., 1991; Schreurs et al., 2003; Zwaan et al., 2018, 2020; Zwaan and Schreurs, 2023) would allow us to better track viscous deformation, which, as we have shown, plays a significant role in promoting basin inversion.

## 4.4 Comparisons with the North Australian Craton

### 4.4.1 1800–1750 Ma wide rifting and 1750–1710 Ma inversion of the Mt Isa terrane

The basins system of the North Australian Craton span more than 0.5 billion years of Earth history during the accretion and dispersal of the Paleo- to Mesoproterozoic Supercontinent Nuna (Betts et al., 2016; Gibson et al., 2018; Johnson, 2021; Kirscher et al., 2020; Zhang et al., 2012). There are numerous interpreted tectonic drivers for the basin evolution of the North Australian Craton (e.g., rifting *sensu stricto*: O'Dea et al., 1997; strike slip tectonics: Southgate et al., 2001). Several tectonic models agree that this series of basins formed in the overriding plate of one or more convergent plate margins (Scott et al., 2000; Giles et al., 2002; Betts and Giles, 2006; Yang et al., 2019). These basins have stratigraphy that can be correlated, even though they are dispersed across the entire craton over a distance on the order of 1000 km (Figure 12). At least four unconformably bounded Superbasins are resolved spanning 1840–1350 Ma. The oldest Superbasin is the Leichhardt Superbasin (ca. 1800–1740 Ma, Jackson et al., 2000).

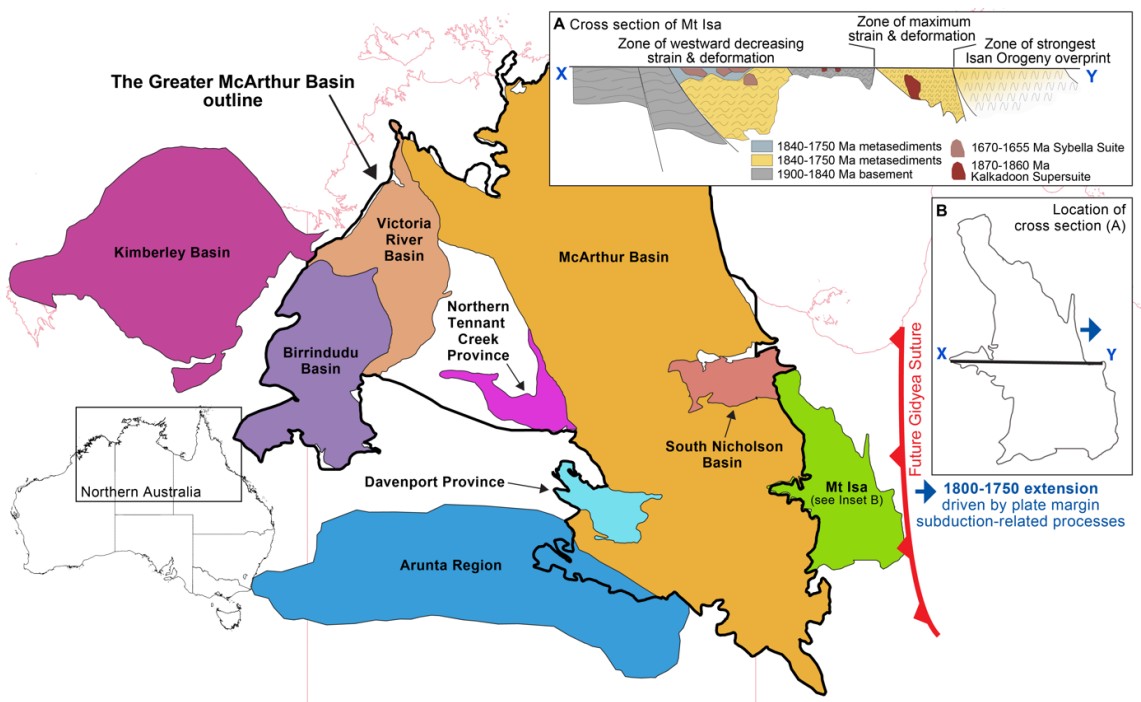

**Figure 12: Map of Proterozoic basins in northern Australia (basin shapes obtained from Geoscience Australia portal: https://portal.ga.gov.au). (a) Cross section of with 1800–1750 Ma showing strain and deformation intensity related to compression during the 1750-1710 Ma Wonga Orogeny (Spence et al., 2022) which followed the 1800-1750 Ma extension. (b) Outline of the Mt Isa terrane, showing the location of (A).**

The Superbasin sequences are usually separated by transient inversion events (Blaikie et al., 2017; Betts, 1999; Spence et al., 2021). Each new Superbasin phase is associated with the renewed reactivation of faults inherited from the previous, underlying basin (e.g., O'Dea et al., 1997b; Betts et al., 2004, 2006). The intensity of the inversion events varies across the North Australian Craton. Inversion is subtle in the interior of the craton (Bull and Rogers, 1996), whereas along the craton edges (e.g., Mount Isa) it is much stronger and dominates the structural grain of the region.

The ~2000 km-long, roughly north-south trending Mount Isa terrane lies in the eastern part of the North Australian Craton (Figure 12). This polydeformed terrane has been affected by multiple extensional and compressional episodes due to its interaction with Laurentia to the east between ~1800 and 1500 Ma (Betts et al., 2006; Betts et al., 2008; Gibson et al., 2018; Korsch et al., 2012; Olierook et al., 2022). The Leichhardt Superbasin is the oldest basin exposed in the region, cropping out throughout the entire Mount Isa terrane (Gibson and Edwards, 2020; Neumann et al., 2006). A ~20 Myr period of extensional activity was followed by a compressional event known as the Wonga Orogeny (Spence et al., 2022), possibly related to the accretion of the seismically-imaged Numil province to the east (Blaikie et al., 2017). The Wonga Orogen appears to decrease in intensity westwards (Figure 12a). The orogen has mostly been recognised in the eastern zone of the Mount Isa region within

the Mary Kathleen Domain. Evidence of this compressional event becomes more scarce to the west, possibly due to overprinting by younger events and lack of exposure (Spence et al., 2022).

The experiments we presented here simulate one-sided extension and shortening, similar to continental extension and shortening resulting from far-field plate margin processes at the eastern margin of the North Australian Craton, i.e., Mount Isa. The wide distribution of rift basins in our experiments is analogous to the distributed basins system of the North Australian craton. During the extensional phase, Models R3, R4, and R5 exhibit partitioning of strain into so-called low-strain zones (characterised by the absence of basins) and high-strain zones (populated by clusters of basins) (Figure 4 and Figure 5). The northern half of these models are dominated by low-strain zones, and the southern half by high-strain zones. During the shortening phase of our experiments, more strain is accommodated in the southern half of the models. We interpret that this strain partitioning is partly due to the displacement gradient imposed by the moving U-shaped wall at the southern end of the model (Figure 2). This displacement (and velocity) gradient and the resulting high strain in the southern part of our models is analogous to the stronger effects of plate margin processes (i.e., west-dipping slab roll-back; Betts et al., 2016), and therefore the intensity of deformation, towards the east of the Mount Isa region (Spence et al., 2022) (Figure 12a).

### 4.4.2 Selective basin inversion: Implications for metamorphism and Pb-Zn mineral systems in Mount Isa

The structures and the metamorphic facies distribution in the Mount Isa region predominantly reflect peak metamorphism during with the Isan Orogeny (e.g., Betts et al., 2006; Foster and Rubenach, 2006; Austin and Blenkinsop, 2008; Blenkinsop et al., 2008). This episode of metamorphism is associated with regional shortening which followed basin-forming rifting and the subsequent thermal sag phase (O'Dea et al., 1997a). The map-view pattern of the metamorphic facies distribution is characterised by north-south trending, amphibolite facies belts separated by zones of mainly greenschist facies rocks (Foster and Rubenach, 2006). This juxtaposition of high- and low-grade metamorphic rocks reflects steep upper crustal thermal gradients, the cause of which has been the subject of debate. McLaren et al. (1999) proposed that the source of heat contributing to the high geothermal gradient is the granitic Sybella batholith that was emplaced at shallow crustal levels during the initial basin-forming rift phase (O'Dea et al., 1997b). They further proposed that high-temperature metamorphism was facilitated by the trapping of heat (from the granitic batholith) within the upper crust by the insulating, overlying basin sediments during protracted rift-related subsidence. Their model takes into account that despite the spatial correlation between this granite batholith and high metamorphic-grade rocks, granite emplacement and peak regional metamorphism are separated by ~130 Myr (Connors and Page, 1995).

The results of our experiments suggest that steep thermal gradients in basin inversion settings could be attributed to: (1) strain localisation by the rift basins during extension prior to basin inversion and (2) the selective inversion of basins during subsequent shortening (Models R3, R4, and R5). In nature, rift basins correspond to areas of crustal thinning, with which high geothermal gradients are associated; areas of active rifting correspond to high heat flow (e.g., Lysak, 1987). During the shortening phase of our experiments, strain is localised at the rift basins, through the reverse reactivation of normal faults and folding of the ductile layers underneath the basins (Figure 7). This interpretation of our experiments, which involved a pause

between extension and shortening that scales to ≤1.7 Myr in nature, is consistent with numerical models which demonstrate that basins remain mechanically weaker when shortening occurs after a short (<25 Myr) post-rift phase (Buiter et al., 2009). By the end of shortening, inverted basins correspond to high topography, while the basins that were not inverted remained as topographic lows. In nature, these low-topography areas would correspond to deeper units that are subjected to higher-temperature (i.e., amphibolite-facies) metamorphism during shortening. In contrast, high-topography areas (i.e., inverted basins) would be subjected to lower-temperature (i.e., greenschist facies) metamorphism. Our experimental observations align with: (1) field observations of the alternating high- and low-grade pattern of metamorphic facies at Mount Isa and (2) the interpretation that this metamorphism is associated with regional shortening. While our isothermal analogue experiments do not directly account for the effects of and changes in temperature during extension and shortening, they provide some insight into the role of the rheological stratification (and by proxy thermal stratification) of the lithosphere during wide rifting and subsequent inversion. More complex future experiments could be designed to investigate the role of post-rift sedimentation in potentially suppressing basin inversion and provide a useful comparison to existing numerical models (e.g., Buiter et al., 2009). Early models of Pb-Zn ore formation at Mount Isa suggested that mineralisation occurred during basin formation and was facilitated by fluid transport along active normal faults (Betts et al., 2003; Betts and Lister, 2001). However, more recent interpretations suggest that basin inversion strongly controlled the emplacement of Pb-Zn mineral systems as well as the development of petroleum systems in Mount Isa (Gibson and Edwards, 2020; Gibson et al., 2017). For the ca. 1575 Ma Century Pb-Zn deposit, Gibson and Edwards (2020) proposed that hydrocarbons and then a more metalliferous ore-forming fluid were consecutively trapped following their ejection from deeper stratigraphic levels during the 1620–1500 Ma Isan Orogeny shortening. They further suggested that Pb-Zn exploration strategies in this region should take into account the close temporal and spatial links between mineral and petroleum systems, the latter of which may consist of hydrocarbon traps associated with inversion structures (e.g., Turner and Williams, 2004).

While previous studies have shown how selective fault reactivation contributes to mineralisation (e.g., Sibson, 1995; Nortje et al., 2011), there has been little focus on the selective inversion of entire basins. Understanding the factors contributing to varying amounts of inversion (between basins) within the same basins system could assist in exploring for ore deposits that formed during pre-inversion extension. The amount of inversion has implications for orebody preservation, as uplifted areas are subject to erosion. For example, many of the Pb-Zn deposits in the Mesozoic basins of western Europe (i.e., France, Spain) formed during extension and then experienced inversion during the Alpine orogeny (Munoz et al., 2016). The young age of the inversion allowed for the preservation and therefore extensive exploration of these mineral systems. Similarly in northern Australia, the extended post-orogenic evolution of the Mount Isa Inlier is characterised by heterogeneous but regionally consistent slow cooling and exhumation (< ~0.5 mm/yr) driven mostly by diachronous fault movements (Li et al., 2020). If the ore deposits in this region had formed during pre-Isan extension (e.g., Betts and Lister, 2001; Betts et al., 2003), this slow uplift could have contributed to ore preservation. Further investigation into the spatial and temporal relationships between basin inversion and mineralisation, as well as the drivers of variable basin inversion in mineralised regions, could provide useful insights for future exploration programs.

**5 Conclusions**

The initial objective of this experimental series was to identify a suitable reference experiment for future, more complex experiments that will investigate multi-stage tectonics in the North Australian Craton. All of the experiments presented here successfully demonstrated wide rifting during the extensional phase, consistent with the formation of a distributed system of basins in the North Australian Craton in the Proterozoic. However, the setup for Models R4 and R5 would be the best candidate for a reference experiment, given that the layer properties and corresponding strength profile are most consistent with previous three-layer models of wide rifting and estimates for the density structure of the natural lithosphere. The relatively simple models of (selective) basin inversion described here already provide additional insights on how deformation was partitioned and how steep metamorphic gradients were formed in the Mount Isa region of the North Australian Craton. We also suggest that developing better constraints on the temporal and spatial relationships between basin inversion and mineralisation could be useful for exploring for Pb-Zn deposits in this region, given that basin inversion impacts orebody preservation.

The analogue experiments presented here demonstrate that basin inversion is driven by deep processes occurring beneath the brittle upper crust. Basin uplift is facilitated by upward movement of the ductile lower crust or mantle. For a distributed system of basins, comparable to a series of basins in a natural wide rift setting, it is possible that only some basins are inverted while others remain as topographic depressions. Selective basin inversion could be related to the superposition of crustal-scale and lithospheric-scale boudinage formed during a previous basin-forming extensional phase or folding of the ductile layers during shortening. These viscous processes occur at a different scale to the reverse reactivation of upper crustal normal faults, which is a frictional process, and may be equally important for driving basin inversion. Cross sectional or 3D imaging of the evolution of basins in analogue experiments, facilitated by non-destructive monitoring methods, could help us better understand the interplay between crustal- and lithospheric-scale structures in facilitating or suppressing basin inversion.

**Data availability**

QGIS project files containing the interpreted fault traces and basins, as well as time series of orthorectified top-view images, strain and displacement maps, and topographic profiles are provided in Samsu et al. (2023): https://doi.org/10.5880/fidgeo.2023.022.

**Author contributions**

AS, WG, PB, and ARC contributed to the conceptualisation of the experiments. ARC, WG and PB acquired funding for this project and its publication. AS conducted the investigation with the assistance of FA and EM. AS, TCS, and EM analysed and visualised the data. The original draft was written by AS, WG, and TCS. All authors contributed to reviewing and editing the manuscript.

## Competing interests

The contact author has declared that none of the authors has any competing interests.

## Acknowledgements

This work was made possible by the Australian Research Council Linkage Grant LP190100146 and MRIWA project M554. We thank the participants of the above projects for discussions related to the experiments reported here. Uchitha Nissanka Arachchige is thanked for assistance in the lab. We also thank Stefan M. Schmalholz for helpful discussions on lithospheric folding, as well as editor Ernst Willingshofer, whose constructive comments improved the manuscript. FA was supported by a Monash University Faculty of Science Dean's Postgraduate Research Scholarship. Support for EM came from the Monash University Faculty of Science 2022 Advancing Women's Success Grant that was awarded to AS.

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
