# Peer review of "Selective inversion of rift basins in lithospheric-scale analogue experiments"

_EGUsphere, 2023_

## Author Response (AR1)

**Response to Reviewer 1's comments**

Note from authors: Our responses to each comment are provided as bullet points. Line numbers of the reviewers' comments refer to the previous version of the manuscript, while line numbers in our responses refer to our revised manuscript.

This paper discusses the results of analogue experiments that simulate basin inversion after extension. The focus is on the behavior of the ductile lower crust when basins invert. The authors find that depending on the strength (rheology) of the ductile layer and mantle basins form in a wide rift setting and are only all inverted when initial basin distribution was evenly spaced. In case of localized extension and shortening only some basins invert and become uplifted. This uplift then occurs at regular intervals.

The focus of this study on the behavior of the ductile lower crust in basin inversion helps us to understand the dynamics behind basin inversion at crustal scale. The methods used are well-established. The focus of their study is unique and the conclusions reached are substantial and lead to more detailed questions beyond the scope of this paper, for example: what influences the wavelength of basin formation and basin inversion?

I think the paper is well-written and well-structured and it advances our understanding of how the ductile lower crust behaves under changing tectonic stress fields. I would like to congratulate the authors with their work and I recommend this work for publication, with only some technical corrections.

With kind regards,

Anouk Beniest

Assistant Professor, Vrije Universiteit Amsterdam

**Technical corrections:**

In the caption of figure 1 you refer to the liquid asthenosphere as 'a fluid asthenosphere'. I suggest to replace 'fluid' by 'liquid' (i.e. 'a liquid asthenosphere'), also in other parts of the text: liquid describes a state of matter ('liquid [state] asthenosphere [matter]'), whereas a fluid refers to any substance that flows.

> ▷ "Fluid" has been replaced by "liquid" in the Figure 1 caption and throughout the text.

In figure 2, the red arrows that indicate the direction of movement are placed parallel along the long side of the model box. I was a bit confused by that, thinking that extension/shortening was imposed on the side walls only. Maybe consider replacing the arrow left or right to the moving arm in the picture, on perpendicular to the short side of the model box.

▷ We now have one arrow for each picture, perpendicular to the short side of the model box.

Line 225: a graben is defined as a 'depressed area'. I understand what you mean, but it reads as if the graben is very sad. Maybe just write 'depressions [or topographic depressions] that are bounded by parallel normal faults'.

▷ That's a great point! We do not want our grabens to be sad. "Depressed area" has been replaced with "topographic depression" [**Line 246**].

Line 256: 'Fault traces are smoother…'What exactly do you mean with smoother? Are they continuous? Less edgy? Longer/shorter? Can you be more specific/descriptive?

▷ We have rewritten this sentence to be more descriptive here [**Lines 278-280**]: "Fault traces in Models R3, R4, and R5 are more clearly defined than in Models R1 and R2, due to the lower quality of PIV data for Models R1 and R2 (possibly caused by sparser distribution of tracking particles on the model surface and non-optimal lighting conditions)."

Line 260: you mention de words 'northernmost' and 'southernmost'. Can you maybe indicate what's north and south to you on the figure? Or maybe use top vs bottom, or use some other reference indicator (A - A', B - B', or something else)?

▷ In **Lines 240-241**, we have described how we orient ourselves when describing the modelling results: "When viewing the models in map view, the upper/top side of the image is referred to as "north", and the model is being extended towards the "south" (bottom side of the image).
▷ North arrows have been added to Figures 4 to 10.
▷ "N" for north and "S" for south have been added to the profiles in Figure 11.

Line 383: '… inversion is localized in the center of the basin…' It is not entirely clear to me if you observe as well that, despite your models being isothermal, the inversion localizes at the center of the basin. Or are you just making the point here that, despite your model being isothermal, the inverted basins are also thinnest in center of the basin, possibly due to isostatic readjustment, hence this would correspond to a higher heat flow? Is the expected higher heat flow in that case a result of the large wavelength thinning of the lithosphere and isostatic equilibration of the thinned regions, or do you still think that the thermal structure is of importance for the location of basin inversion? I think this is a very interesting point that you are making, so could you maybe elaborate a bit more on the consequences of lithosphere thinning on the potential changes in heat flow?

▷ We have rewritten this segment to better reflect our ideas on the feedback between lithospheric thinning and thermal weakening, as follows [**Lines 418-427**]: "In their thermo-mechanical experiments of basin inversion, Sandiford et al. (2006) found that inversion is localised in the centre of the basin due to higher than average thermal gradients beneath the basin centre. In our analogue models, we observed that inversion is also greatest in the basin centre, based on maps of

cumulative vertical displacement (Figure 8). Even though we cannot directly observe the influence of heat flow in our isothermal experiments, we speculate that at the end of extension, the basin centres in our models lie above thinned parts of the model lithosphere. In nature, lithospheric necking corresponds to a reduction in rift strength as hotter asthenospheric mantle material replaces colder and stronger lithospheric mantle (Chenin et al., 2018). In our models, lithospheric necking (Section 4.1) allows the weaker model asthenosphere to replace stronger lithosphere due to isostasy. Combining the results of Sandiford et al. (2006) and our experiments, we suggest that basin inversion in nature occurs where the lithosphere is weakest, and this zone of weakness is created by thinning of the strongest layers of the lithosphere and the upwelling of hot asthenospheric mantle material."

**Response to Reviewer 2's comments**

Note from authors: Our responses to each comment are provided as bullet points. Line numbers of the reviewers' comments refer to the previous version of the manuscript, while line numbers in our responses refer to our revised manuscript.

The Authors present lithospheric scale analogue models to analyze the process of rift basin inversion during shortening. To this aim they deform the models in extension first, followed by later compression and analyze the role of different parameters (e.g., layer thickness, rate of extension) on the process.

In found the paper interesting, as it represents of the few examples of analogue modelling of lithospheric scale inversion processes. The results are also very interesting and useful for a wide audience. However, I found some major problems in the model setup/scaling/description of the series that should be fixed; they are summarized below.

**Main comments**

The main problem I found with this paper is that it is -at least in some parts- not always clear.

For instance, it is not very easy to understand what parameters are systematically changed in the 5 models; some differences are mentioned here and there in sections 2.1 and 2.2 but I think the experimental series (i.e., what are the main differences among the models and why these parameters have been changed) should be clearly described in (a separate subparagraph of?) section 2.

▷ We added an introductory paragraph for Section 2, where we explain the initial objective of the experimental series and the main differences between Models R1/R2 and Models R3/R4/R5 [**Lines 95-102**]: "The experimental setup for extension followed by shortening is illustrated in Figure 2. The initial objective of the experiments presented here was to identify a suitable reference experiment of wide rifting and subsequent shortening, against which future experiments (e.g., those that include pre-existing weaknesses) can be compared. Hence, multiple parameters were changed between experiments (Table 1). Models R1 and R2 consisted of an upper crust that was very thin relative to the ductile lower crust. In contrast, the thickness ratio between the upper and lower crust in Models R3, R4, and R5 was 50:50, which is more representative of the North Australian Craton (Section 2.1). The extension and shortening velocity for models R1 and R2 was also much slower than for R3, R4, and R5. As a result, the lithospheric mantle in R1 and R2 is weak compared to R3, R4, and R5 (Figure 3), resulting in differences in the strain localisation behaviour of the lithospheric mantle (Section 3)."

Related to this, in the models (R3-R5), the Authors have changed both the velocity and the thickness of both the UC and LC (but also the rheology of some layers) with respect to models R1-R2, i.e. more than one parameter is changed in each model with respect to

the standard model. This is not strictly correct for a parametric analysis and should be somehow explained and justified.

▷ The response to the previous comment addresses the change of the velocity and thicknesses of the crustal layers.

▷ The change in rheology between the models is mainly due to the change in effective viscosity of the ductile layers as a function of the extension/shortening velocity. We discuss this effect in **Lines 163-168**: "However, in Models R1, R2, and R3, the use of ultrafine-grained iron filings (manufactured for Mad About Science), which has a powder-like consistency, had the unintended effect of doubling the viscosity of the PDMS-based mixture (compare LC1 and LC2 in Table 2). We opted to use fine-grained iron filings for Models R4 and R5 to mitigate this viscosity increase. Hence the yield strength profile for Model R3 contains a "strong" lower crust, while Models R4 and R5 contain a lower crust that is significantly weaker than the lithospheric mantle (Figure 3); the latter is more consistent with theoretical strength profiles for a wide rift setting."

▷ In the new introductory paragraph for Section 2, we imply that this study is not a parametric analysis, but instead an effort to find a suitable reference experiment for future parametric studies (see response to previous comment).

Also, some details of the scaling of models are anticipated in section 2.1, then a detailed description of the procedure is described in section 2.2 (but also here some details are anticipated at the beginning and then addressed in detail later – see specific points below).

▷ As noted by the reviewer, it is correct that some details on scaling are introduced in Section 2.1, whereby more details are included in Section 2.2.

▷ In Section 2.1, we include some details on scaling to give the reader a sense of how the boundary conditions of the model (i.e., thicknesses and velocity), compare to the Basin and Range Province and North Australian Craton natural examples, the latter of which is the inspiration for this experimental series [**Lines 112-123**]: "As the experiments were designed to help us better understand Proterozoic craton-wide rifting in the North Australian Craton (Allen et al., 2015), we implemented a rheological layering that allowed extension to be relatively uniform across the entire model area and create a distributed system of basins (i.e., "wide rifting" sensu Buck, 1991; also see Brun, 1999 and Buck et al. 1999). Hence the model lithosphere is analogous to a natural thick lithosphere (with a thick crust) shortly after orogenesis or with a higher-than-normal heat flow (Buck et al., 1999). In Models R1 and R2, the thicknesses of the crustal layers scale to 10 km and 40 km for the upper and lower crust, respectively; these were modelled after crustal thickness estimates for the Basin and Range Province (Gueydan et al., 2008), a well-known example of a wide rift (e.g., Hamilton, 1987; Parsons, 2006). The upper and lower crust layers in Models R3, R4, and R5 have the same thicknesses, which is representative of the North Australian Craton (Betts et al., 2002; Kennett et al., 2011)…"

▷ In Section 2.2, we describe in more detail how the scaling factors were chosen [**Lines 169-201**].

As for the scaling, in section 2.2 many parameters are scaled based on the scaling factors of the asthenosphere. This seems to be quite strange since it is the less important among the ductile layers. Also strange is the assumption of the difference of the natural value of the asthenosphere in the difference models – why this? This is not explained and I think (as stated above) that this (and the description of the difference among the different models explained above) creates a lot of confusion. To be honest I got lost at some point reading this paragraph.

▷ As noted in the text, the scaling factors for density and viscosity were based on the asthenosphere, following Schellart (2011) and previous brittle-ductile analogue experiments that included a liquid asthenosphere (e.g., Molnar et al., 2017; Santimano and Pysklywec, 2020; Samsu et al., 2021) [**Lines 174-177**].

▷ The paragraph mentioned by the reviewer has been rewritten. We added a brief explanation on the necessary changes to the scaling parameters as we progressed from Models R1 and R2, to R3, and finally to R4 and R5 [**Lines 183-191**]: "For Models R1 and R2, we started out with a natural asthenosphere density $\rho_p$ = 3,100 kg/m$^3$ and viscosity $\eta_p$ = 1.9 x 10$^{19}$ Pa s, as we planned an extension duration of 14 hours. For Model R3, we wanted to explore what happens when we extended the model by the same amount but at a faster rate (in around three hours). Therefore, it was necessary to increase the prototype viscosity by one order of magnitude (to $\eta_p$ = 1.9 x 10$^{20}$ Pa s) to achieve an appropriate time scaling factor. Finally, for Models R4 and R5, we created an improved lithospheric mantle mixture (LM2 in Table 2) with the desired viscosity $\eta_m$ = 2.7 x 10$^5$ Pa s (approximately ten times greater than the model lower crust). As this mixture had a density $\rho_m$ = 1,384 kg/m$^3$, the density scaling factor was changed (using $\rho_p$ = 3,400 kg/m$^3$ for the asthenosphere), otherwise the prototype lithospheric mantle and asthenosphere densities would have both equalled 3,100 kg/m$^3$. This last change did not significantly impact the other scaling factors."

Discussion – section 4.1. This part is rather long and may be somehow shortened also given that it is not always clear. For instance, in lines 339-343, how can the asthenosphere influence the distribution and characteristics of deformation in the overlying brittle-ductile multilayer is not very clear to me. The rheological characteristics (which also depend on the strain rate) and thickness of the brittle/ductile lithospheric layers have been suggested to play a major role in controlling deformation. This has to be clarified. Also the comparison with some previous experiments (e.g., Benes and Davy, 1996 in lines 345-347) is simply described but not discussed. Lines 366-368: how this process occurs is not very clear to me, and should be maybe clarified.

▷ The comparison with Benes and Davy (1996) has been expanded upon [**Lines 366-369**]: "In Models R3, R4, and R5, basin spacing is on the order of the initial crustal thickness (18 mm). Benes and Davy (1996) observed a similar relationship between basin spacing and crustal thickness in their wide rift analogue

experiments, from which they interpreted that the characteristic wavelength of crustal-scale periodic instabilities is on the order of the crustal thickness."

▷ The comparison with analytical models (e.g., Fletcher and Hallet, 1983) has been made clearer [**Lines 358-365**]: "In Models R3 and R4, zones of localised lithospheric necking may correspond with high-strain zones, while areas that underwent minimal stretching may correspond with low-strain zones (Figure 5 and Figure 6). Here, the distances between the centres of high-strain zones may represent the characteristic wavelength of lithospheric-scale boudinage. In Model R4, the spacing between high strain zones is approximately 110 mm. The initial thickness of the ductile layers is 20 mm, giving a dominant wavelength/thickness ratio of 5.5, which is higher than the analytically predicted ratio of ~4 for lithospheric necking (Smith, 1977; Fletcher and Hallet, 1983). This disagreement may partly be due to the low stress exponent of our ductile layers (n $\leq$ 1.4) and the influence of the overlying upper crustal sand layer."

▷ As this paragraph has been heavily rewritten, mentioning how the asthenosphere may have influenced the distribution and characteristics of deformation in the overlying brittle-ductile layer became no longer relevant and has been removed.

▷ "The rheological characteristics (which also depend on the strain rate) and thickness of the brittle/ductile lithospheric layers have been suggested to play a major role in controlling deformation": It is not clear what part of Section 4.1 this comment refers to.

Discussion – section 4.2. This work lacks a detailed analysis of internal deformation, i.e. cross-sections of the models. In fact, some important interpretations are based on topography analysis or observation of ductile crust deformation (after removal of the UC), and there is no direct observation of some important aspects such as (for instance) normal fault reactivation that could directly observed by cutting sections. This is in part understandable, given the difficulty to cut cross-sections of these lithospheric scale models but I think it is should be clearly introduced and discussed in this section (Fig. 12 is a conceptualization of cross-sectional deformation, which is however not directly observed).

▷ We appreciate that the lack of cross-sections (which are challenging to make for this type of three-layer model lithosphere resting on a sticky fluid) hinders direct observations of important processes that are invoked in this manuscript, such as lithospheric boudinage and folding. We have discussed this limitation and suggest ways to image internal deformation in the future, citing the pioneering work of Zwaan and Schreurs (2023) which documents the use of CT-scanning to image lithospheric necking and the upwelling of viscous mantle material during extension [**Lines 482-486**]: "The lack of model cross sections in this investigation, which are challenging to make for a three-layer model lithosphere resting on glucose, hinders direct observations of important processes that are invoked, such as lithospheric boudinage and the upwelling of ductile and liquid material. Imaging of the ductile layers during experimental runs (e.g., using a CT-scanner; Zwaan and Schreurs, 2023) would allow us to better track viscous deformation, which, as we have shown, plays a significant role in promoting basin inversion."

▷ Even though folding could not be directly observed in cross section, we carried out simple calculations which found that folding is theoretically possible for Models R3, R4, and R5 [**Lines 454-468**]: "While it is outside the scope of this work to analyse in detail how $\lambda_v$ is influenced by the model setup (e.g., layer thicknesses, viscosity ratios, bulk strain rates), we introduce here some simple calculations of whether folding of the ductile lithosphere layers is plausible. By treating the combined ductile lower crust and lithospheric mantle as a single ductile layer resting on a homogeneous viscous medium, we can determine an Argand number $Ar \approx 3$ for Models R3, R4, and R5, using the following equation (Schmalholz et al., 2002)…"

▷ We think that the conceptualisation of cross-sectional deformation (Figure 12) is important for (1) summarizing our results and inferences in a conceptual way, and to (2) stimulate future discussions on how such lithospheric-scale analogue experiments could be improved, given that they have the potential to reveal the internal deformation and dynamics of rifts and inverted basin systems. We have made it clear in the caption of Figure 12 that this conceptualisation is partly based on inferences [**Lines 408-411**]: "Upper crustal deformation is observed directly from photographs and PIV-derived topographic and strain maps. Deformation of the ductile and liquid layers are inferred from observations of the top of the lower crust after the upper crust has been removed and are also inspired by previous analogue experiments (Benes and Davy, 1996; Gartrell, 1997)."

▷ Despite us not being able to "see" normal fault reactivation in cross section, the maps of axial strain clearly show high strain along basin-bounding faults during the early stages of shortening (Figures 7 and 8). We mentioned this observation of reverse reactivation of normal faults in the Results [**Lines 290-293**] and Discussion [**Lines 393-397**] sections.

Discussion – section 4.2. Lines 400 and following. These are very interesting findings and I think the Authors could try to analyse (or at least describe) them is some more detail. For instance, there is no mention of mechanical brittle/ductile coupling in these lines (as done in section 4.1) and the Authors only discuss the distribution of deformation in terms of fast/slow extension. Brittle/ductile coupling should be introduced (and maybe calculated?) here.

▷ Thank you for this comment. We expanded the discussion on mechanical coupling between brittle and ductile layers as follows [**Lines 438-443**]: "In this model, the brittle upper crust was much stronger than the underlying ductile layers, so that basin formation was decoupled from uniform extension in the underlying ductile layers (see Section 4.1 for an extended discussion on mechanical coupling between the model layers). Similarly, upper crustal deformation during shortening of Model R2 may have been driven by the reverse reactivation of basin-bounding faults, but here there may have also been the significant influence of upwelling of ductile material underneath thinned crust, unrelated to the mechanical coupling between the brittle and ductile layers."

▷ We have limited our discussion of brittle-ductile coupling to the extension stage of our experiments, as our findings suggest that basin inversion is driven by the upwelling of viscously deforming material, which is driven by isostasy and

influenced by crustal to lithospheric-scale weaknesses in the previously extended model, and/or folding during shortening [**Sections 4.2 and 4.3**]. It is not clear what the reviewer means by calculating brittle-ductile coupling. The brittle-ductile thickness ratio for each model is provided in Table 1.

**Specific comments**

Line 18. not very clear to me.

▷ It is not clear what aspect of Line 18 is unclear, but we have edited a few sentences to hopefully make them clearer [**Lines 17-20**]: "When deformation in the ductile layers is localised during extension (i.e., necking) and shortening (i.e., folding), only some basins – which are evenly spaced apart – are inverted. We interpret the latter as selective basin inversion, which may be related to the superposition of crustal-scale and lithospheric-scale boudinage during the previous basin-forming extensional phase and/or folding of the ductile layers during shortening."

Lines 33-34. 'Zwaan et al....of subsidence' is this sentence necessary? I think it can be removed, since the definition of positive inversion is introduced in the following sentence.

▷ We have removed the sentence beginning with "Zwaan et al...", as per the reviewer's suggestion.

Line 37. Remove ')' after '2022'.

▷ Done.

Line 103. The concept of brittle-ductile crust is already introduced; so this is a pretention that can be removed.

▷ We have kept this part as is, first presenting the brittle and ductile characteristics of the model layers and then specifically comparing the model's strength profiles to natural lithospheric strength profiles.

Lines 108-113. Models R1 to R5 should be described better in a subsection, indicating all the parameters changed (listed in table 1).

▷ Repeated response to previous comment: We added a brief explanation on the necessary changes to the scaling parameters as we progressed from Models R1 and R2, to R3, and finally to R4 and R5 [**Lines 183-191**]: "For Models R1 and R2, we started out with a natural asthenosphere density $\rho_p$ = 3,100 kg/m$^3$ and viscosity $\eta_p$ = 1.9 x 10$^{19}$ Pa s, as we planned an extension duration of 14 hours. For Model R3, we wanted to explore what happens when we extended the model by the same amount but at a faster rate (in around three hours). Therefore, it was necessary to increase the prototype viscosity by one order of magnitude (to $\eta_p$ = 1.9 x 10$^{20}$ Pa s) to achieve an appropriate time scaling factor. Finally, for Models R4 and R5, we created an improved lithospheric mantle mixture (LM2 in Table 2) with the desired

viscosity $\eta_m$ = 2.7 x 10$^5$ Pa s (approximately ten times greater than the model lower crust). As this mixture had a density $\rho_m$ = 1,384 kg/m$^3$, the density scaling factor was changed (using $\rho_p$ = 3,400 kg/m$^3$ for the asthenosphere), otherwise the prototype lithospheric mantle and asthenosphere densities would have both equalled 3,100 kg/m$^3$. This last change did not significantly impact the other scaling factors."

Line 154. Scaling of velocity anticipated here and then presented later.

▷ As mentioned in the response to a previous comment, we include some details on scaling in Section 2.1 to give the reader a sense of how the boundary conditions of the model (i.e., thicknesses and velocity) compare to natural examples. For example, the reviewer is referring to a sentence in which we justify the scaled rate of extension using the Basin and Range Province (a classic example of a wide rift) as a natural analogue [**Lines 121-123**]. In Section 2.2, we describe in more detail how the scaling factors were chosen [**from Line 169**].

Line 158 and following. See above.

▷ Repeated response to previous comment: We added a brief explanation on the necessary changes to the scaling parameters as we progressed from Models R1 and R2, to R3, and finally to R4 and R5 [**Lines 183-191**]: "For Models R1 and R2, we started out with a natural asthenosphere density $\rho_p$ = 3,100 kg/m$^3$ and viscosity $\eta_p$ = 1.9 x 10$^{19}$ Pa s, as we planned an extension duration of 14 hours. For Model R3, we wanted to explore what happens when we extended the model by the same amount but at a faster rate (in around three hours). Therefore, it was necessary to increase the prototype viscosity by one order of magnitude (to $\eta_p$ = 1.9 x 10$^{20}$ Pa s) to achieve an appropriate time scaling factor. Finally, for Models R4 and R5, we created an improved lithospheric mantle mixture (LM2 in Table 2) with the desired viscosity $\eta_m$ = 2.7 x 10$^5$ Pa s (approximately ten times greater than the model lower crust). As this mixture had a density $\rho_m$ = 1,384 kg/m$^3$, the density scaling factor was changed (using $\rho_p$ = 3,400 kg/m$^3$ for the asthenosphere), otherwise the prototype lithospheric mantle and asthenosphere densities would have both equalled 3,100 kg/m$^3$. This last change did not significantly impact the other scaling factors."

Lines 230-231. 'In general, the rift basins…' This is not very clear to me. Do the Authors mean something like 'the basins were fully established when …'?

▷ This sentence has been rewritten as [**Lines 252-253**]: "Basins became fully established when the normal basin-bounding faults reached their final length."

Conclusions. Lines 531 and following. These are mostly technical aspects, some of which are (to me) not worth of a description here (e.g., the role of isostasy). I would mention something related to the natural example instead.

▷ In the rewritten Conclusions section, some technical aspects (e.g., the role of isostasy) have been removed, as suggested by the reviewer.

▷ We have added a paragraph that links our experiments to the North Australian Craton natural case study [**Lines 583-590**]: "One of the main objectives of this experimental series was to identify a suitable reference experiment for future, more complex experiments that will investigate multi-stage tectonics in the North Australian Craton. A setup like that used for Models R4 and R5 would be the best candidate for such a reference experiment, given that the corresponding strength profile is most consistent with estimated lithospheric strength profiles for wide rift settings in nature. Nevertheless, the relatively simple models of (selective) basin inversion described here already provide some insights on how deformation was partitioned and how steep metamorphic gradients were formed in the Mount Isa region of the North Australian Craton. We also suggest that developing better constraints on the temporal and spatial relationships between basin inversion and mineralisation could be useful for exploring for Pb-Zn deposits in this region, given that basin inversion impacts orebody preservation."

---

## Author Response (AR2)

**Response to Topic Editor's comments**

Note from authors: Our responses to each comment are provided as bullet points. Line numbers of the reviewers' comments refer to the previous version of the manuscript, while line numbers in our responses refer to our revised manuscript.

Dear authors,

Overall the concept of your contribution (inversion of wide-rifts) is interesting and the manuscript has clearly improved with respect to the original submission.

Based on my own reading, some additional clarifications are needed, particularly on the justification of the material properties and rheological setups when compared to nature (section 2). There, a better justification of your choices how the model material properties reflect the properties of natural systems is needed. The natural example (North Australia Craton) should be leading when it comes to the choice of modelling materials and their properties. Here you follow an unclear logic, where you scale the natural example to the model to match the desired time-scales (eg. L185/186). An important implication of your approach is that you assume unrealistically low (eg. 2900 kg/m3 in table 2) densities for the lithospheric mantle which also vary considerably among models, which provides a flavour of arbitrariness, which you wish to avoid. Some comments of reviewer R#2 were along these lines when questioning the parameter space, you explored.

- We thank the editor for these constructive comments. We acknowledge the need for better justifying how the model material properties reflect the properties of the natural lithosphere, where possible with comparisons with the North Australian Craton. We have added justifications for our choice of scaling properties, for example:
  - In **Lines 127-133**, we explain the choice of crustal thicknesses: "Given the challenge of reconstructing the lithosphere configuration and rifting conditions of the North Australian Craton in the Proterozoic, we used the Basin and Range Province – a well-known example of a wide rift (e.g., Hamilton, 1987; Parsons, 2006) – as a proxy for estimating crustal thicknesses (Gueydan et al., 2008) and the rate of extension for our models. Hence, the thicknesses of the crustal layers in Models R1 and R2 scale to 10 km and 40 km for the upper and lower crust, respectively. After running Models R1 and R2, we found that it would be more representative of the North Australian Craton (Betts et al., 2002; Kennett et al., 2011) to have upper and lower crust layers with the same thickness, which we then implemented in Models R3, R4, and R5 (**Error! Reference source not found.**)."
  - In **Lines 196-199**, we clarified that estimated asthenosphere densities between 3100 kg/m$^3$ and 3400 kg/m$^3$ are "consistent with previous lithospheric-scale analogue experiments (e.g., Molnar et al., 2017; Santimano and Pysklywec, 2020; Samsu et al., 2021) and reference asthenospheric densities used in geophysical models (e.g., 3250 kg/m$^3$ in Lamb et al., 2020)."

- In **Lines 217-220**, we explained that the "layer densities in Models R4 and R5 are the most consistent with those used in geophysical models on the density structure of the lithosphere, where the densities of the upper crust, lower crust, and lithospheric mantle are 2700 kg/m$^3$, 2940 kg/m$^3$, and 3350 kg/m$^3$ respectively (Kaban et al., 2014)."
    - Related to the last point, we reiterated in several parts of the text that (1) the initial objective of this experimental series was to identify a reference experiment of wide rifting followed by shortening, to be compared against future, more complex experiments for understanding multistage tectonics in the North Australian Craton [**Lines 97-101**, **634-635**]; and (2) Models R4 and R5 were the most suitable, as "the layer properties and corresponding strength profile are most consistent with previous three-layer models of wide rifting and estimates for the density structure of the natural lithosphere" [**Lines 637-639**].
- We hope that we have provided a clearer explanation on the changes to the scaling parameters between Models R1 and R2, Model R3, and Models R4 and R5 in **lines 205-220**: "For Models R1 and R2, we started out with a natural asthenosphere density $\rho_p$ = 3100 kg/m$^3$ and viscosity $\eta_p$ = 1.9 x 10$^{19}$ (following Molnar et al., 2017) and an extension velocity that scaled to 2 mm/year, resulting in an extension duration of 14 hours. For Model R3, the objective was to explore the behaviour of the ductile layers when we extended the model by the same amount but at a faster rate. Therefore, the prototype viscosity was increased by one order of magnitude (to $_p$ = 1.9 x 10$^{20}$) to achieve an appropriate time scaling factor. This change in the time scaling factor enabled us to apply an extension rate that still scaled to 2 mm/year in nature within a shorter (experimental) extension duration, i.e., around 3 hours. However, additional changes to the ductile materials were still necessary, as the strength contrast between the lower crust and lithospheric mantle (LM1 in Table 2) in Model R3 was too low to simulate natural lithosphere with a strong lithospheric mantle and relatively weak lower crust (Table 1, Figure 2c). Therefore, for Models R4 and R5, we created an improved lithospheric mantle mixture (LM2 in **Error! Reference source not found.**) with the desired viscosity $\eta_m$ = 2.7 x 10$^5$ Pa s (approximately ten times greater than the model lower crust), resulting in a low LC:LM strength ratio. As this mixture had a density $\rho_m$ = 1384 kg/m$^3$, the density scaling factor was changed to 0.42 (using $\rho_p$ = 3400 kg/m$^3$ for the asthenosphere), otherwise the prototype lithospheric mantle and asthenosphere densities would have both equalled 3100 kg/m$^3$. This last change did not significantly impact the other scaling factors. The layer densities in Models R4 and R5 are the most consistent with those used in geophysical models on the density structure of the lithosphere, where the densities of the upper crust, lower crust, and lithospheric mantle are 2700 kg/m$^3$, 2940 kg/m$^3$, and 3350 kg/m$^3$ respectively (Kaban et al., 2014)."
- The choice of an initial lithospheric mantle density of 2900 kg/m$^3$ in Models R1 and R2 (Table 2) was not influenced by the time scaling. Instead, this initial choice was arbitrary, mainly focusing on choosing a prototype lithospheric mantle density that was lower than the asthenosphere to prevent subduction of the model

> lithosphere. We realised after running Models R1 and R2 that changes to the scaling parameters were required (see previous point).

The discussion sections would benefit from integrating some key references. For example, with reference to vertical motions related to folding of a weak basin see Dombradi et al., 2010 (doi:10.1016/j.tecto.2009.09.014). Though this paper does not have an extension phase proper, the model mechanical stratigraphy at the onset of shortening is comparable to your models R1 and R2. Other suggestions are provided in particular for sections 4.1 and 4.2 (for details see the annotated manuscript).

- Key references have been added in various parts of the discussion, e.g., in relation to upwelling of viscous material during extension [**Lines 449-451**] and CT-scanning [**Lines 532-533**].
- We added a sentence relating vertical motions to folding [**Lines 442-447**]: "Continued shortening resulted in inversion of the basins, which we interpret to have been driven by anticlinal folding of the ductile layers, based on observations of uplifted lower crust underneath the inverted basins (following the removal of upper crustal material at the end of the experiments). This interpretation is comparable with observations from analogue experiments of continental collision (Sokoutis and Willingshofer, 2011) and intraplate compression (Dombrádi et al., 2010), where strain is accommodated and topography is controlled by folding of pre-existing weak zones."
- All suggestions for Sections 4.1 and 4.2 in the annotated manuscript have been addressed (see comments in the revised manuscript with tracked changes).

Figures: I suggest to remove fig. 2c and 3b and merge what is left of Fig 2 and 3 into one figure, which shows the fundamentals of the experimental setup and rheology. Make sure that all abbreviations in the figures are explained in the captions.

- We have merged the edited versions of the previous Figures 2 and 3 into one figure (now Figure 2) that shows the fundamentals of the experimental setup and rheology.

Please see the annotated manuscript for detailed comments and suggestions.

- All comments and suggestions in the annotated manuscript have been addressed.
- Editor's comment on what was previously Figure 10 (now Figure 9): "Model R3 seems to have a tilt from upper left to lower right corner. Why? Maybe the data need to be corrected for that?"
  - Our response: The tilt that is visible in the oblique 3D view in DaVis was corrected during the strain processing step outlined in Section 3.2. For example, the tilt is no longer visible in Figure 8.

Looking forward to receiving the revised version of the manuscript.

- We thank the editor for the thorough review of the manuscript.

Ernst Willingshofer (Guest Editor)